# Improved Regret and Contextual Linear Extension for Pandora's Box and Prophet Inequality

**Junyan Liu**[*]
University of Washington
junyanl1@cs.washington.edu

**Ziyun Chen**[*]
University of Washington
ziyuncc@cs.washington.edu

**Kun Wang**[*]
Purdue University
wang5675@purdue.edu

**Haipeng Luo**
University of Southern California
haipengl@usc.edu

**Lillian J. Ratliff**
University of Washington
ratliffl@uw.edu

## Abstract

We study the Pandora's Box problem in an online learning setting with semi-bandit feedback. In each round, the learner sequentially pays to open up to $n$ boxes with unknown reward distributions, observes rewards upon opening, and decides when to stop. The utility of the learner is the maximum observed reward minus the cumulative cost of opened boxes, and the goal is to minimize regret defined as the gap between the cumulative expected utility and that of the optimal policy. We propose a new algorithm that achieves $\widetilde{\mathcal{O}}(\sqrt{nT})$ regret after $T$ rounds, which improves the $\widetilde{\mathcal{O}}(n\sqrt{T})$ bound of Agarwal et al. [2024] and matches the known lower bound up to logarithmic factors. To better capture real-life applications, we then extend our results to a natural but challenging contextual linear setting, where each box's expected reward is linear in some known but time-varying $d$-dimensional context and the noise distribution is fixed over time. We design an algorithm that learns both the linear function and the noise distributions, achieving $\widetilde{\mathcal{O}}(nd\sqrt{T})$ regret. Finally, we show that our techniques also apply to the online Prophet Inequality problem, where the learner must decide immediately whether or not to accept a revealed reward. In both non-contextual and contextual settings, our approach achieves similar improvements and regret bounds.

## 1 Introduction

Pandora's Box is a fundamental problem in stochastic optimization, initiated by Weitzman [1978]. In this problem, the learner is given $n$ boxes, each associated with a known cost and a known reward distribution. The learner may open a box at a time, paying the corresponding cost to observe a realized reward. Based on the known distributions, the learner designs a policy that specifies the order of inspecting boxes and a stopping rule for when to halt the process and select a reward. Weitzman [1978] shows that the optimal policy computes a threshold (i.e., reservation value) for each box based on its cost and distribution, and opens boxes in descending order of these thresholds until a stopping condition is met. This model abstracts a wide range of sequential decision-making scenarios where

---

[*]Equal contribution

39th Conference on Neural Information Processing Systems (NeurIPS 2025).

information is costly to acquire and decisions must be made under uncertainty. For example, in the hiring process, an employer interviews candidates one by one, where each interview incurs a cost (e.g., time), and each candidate's performance is drawn from a known distribution (e.g., candidate's profile). The employer must decide when to stop interviewing and make an offer, balancing the expected benefit of finding a better candidate in the future against the accumulated cost of continued search. Other real-world applications include online shopping and path planning [Atsidakou et al., 2024].

While the classic Pandora's Box problem and its variants have been extensively studied (see survey [Beyhaghi and Cai, 2024]), recent work [Fu and Lin, 2020, Guo et al., 2021, Gatmiry et al., 2024, Agarwal et al., 2024, Atsidakou et al., 2024] has started to explore settings where the reward distributions are *unknown*. This line of work is motivated by practical scenarios where distributions are unavailable upfront and must be learned through interaction. These formulations move beyond the classical setting by requiring the learner to adapt based on observed outcomes. Prior studies have considered probably approximately correct (PAC) guarantees, establishing the near-optimal sample complexity Fu and Lin [2020], Guo et al. [2021], and regret minimization over a $T$-round repeated game Gergatsouli and Tzamos [2022], Gatmiry et al. [2024], Agarwal et al. [2024], Atsidakou et al. [2024]. In the basic regret minimization setting, the learner repeatedly solves a Pandora's Box problem at each round and observes the rewards of the opened boxes (a.k.a. semi-bandit feedback), drawn from *fixed but unknown* distributions. However, even in this setting, a gap remains between the $\Omega(\sqrt{nT})$ lower bound of Gatmiry et al. [2024] and the best known upper bound of $\widetilde{\mathcal{O}}(Un\sqrt{T})$ by Agarwal et al. [2024], where $U$ is the magnitude of the utility function that can be as large as $n$.

In addition to the fixed-distribution setting, prior work studied the online Pandora's Box problems with *time-varying* reward distributions, which better capture many real-world scenarios Gergatsouli and Tzamos [2022], Atsidakou et al. [2024]. For example, in the hiring process, the distribution of candidate quality may shift over time due to market trends or seasonal patterns. To model such dynamics, Gergatsouli and Tzamos [2022] consider a setting where the rewards are chosen by an oblivious adversary and show that sublinear regret is achievable when the algorithm competes against a benchmark weaker than the optimal policy, obtained by imposing constraints on the set of boxes it can open. Later, Gatmiry et al. [2024] showed that in the same setting, when the algorithm competes against the optimal policy, no algorithm can achieve sublinear regret, even with full information feedback (i.e., the learner observes the rewards of all boxes regardless of which policy is played). To make the problem tractable while still allowing distributions to change over time, Atsidakou et al. [2024] propose a contextual model where, in each round, the learner observes a context vector for each box, whose optimal threshold can be approximated by a function of the context and an unknown vector. Their approach reduces the problem to linear-quadratic online regression, which provides generality but leads to a regret bound with poor dependence on the horizon $T$. In particular, if the function is linear, they achieve $\widetilde{\mathcal{O}}(nT^{5/6})$ regret bound with semi-bandit feedback (referred to as bandit feedback in their work).

Motivated by these limitations, we address two main challenges in this paper. First, we close the gap in the regret dependence on the number of boxes $n$. Second, we propose a new contextual model that allows the reward distributions to vary over time, and under this setting, we achieve improved dependence on the time horizon $T$. Our contributions are summarized as follows, and further related work is discussed in Appendix B.

- For the online Pandora's Box problem, we propose a new algorithm that achieves a regret bound of $\widetilde{\mathcal{O}}(\sqrt{nT})$, improving upon the $\widetilde{\mathcal{O}}(Un\sqrt{T})$ bound by Agarwal et al. [2024] and matching the known $\Omega(\sqrt{nT})$ lower bound (up to logarithmic factors) [Gatmiry et al., 2024]. Our algorithm builds on the optimism-based framework of Agarwal et al. [2024], but introduces a simple yet crucial modification. Specifically, we first construct empirical distributions for each box using the observed reward samples, and then shift probability mass to form optimistic distributions that encourage exploration. In contrast to the fixed-mass shifting scheme used in Agarwal et al. [2024], we adaptively reallocate mass at each point based on problem-dependent factors. As we demonstrate below, this adaptive construction is central to enabling a more refined regret analysis.

- We extend these results to the contextual linear setting, where each box's expected reward is linear in some known but time-varying $d$-dimensional context and the noise distribution is fixed over time. Our algorithm preserves the overall structure of the non-contextual case, but modifies the construction of optimistic distributions to account for context-dependent variation. Since the reward

samples are no longer identically distributed, we adjust not only the probability mass but also the observed rewards themselves in a *value-optimistic* manner to maintain optimism. While the algorithm of Atsidakou et al. [2024] applies to our setting with $\widetilde{\mathcal{O}}(nT^{5/6})$ regret (see Appendix B), we establish $\widetilde{\mathcal{O}}(nd\sqrt{T})$ regret. As long as $d = o(T^{1/3})$, our algorithm yields a better regret guarantee.

- Finally, we extend our algorithms and analytical techniques to the Prophet Inequality problem in both non-contextual and contextual settings. In the non-contextual case, our algorithm achieves $\widetilde{\mathcal{O}}(\sqrt{nT})$ regret bound, which again improves upon the $\widetilde{\mathcal{O}}(n\sqrt{T})$ regret bound of Agarwal et al. [2024]. For the contextual linear setup, we also establish a $\widetilde{\mathcal{O}}(nd\sqrt{T})$ regret bound. Due to space constraints, all results and proofs for the Prophet Inequality problem are deferred to the appendix.

**Technical Overview for $\widetilde{\mathcal{O}}(\sqrt{nT})$ Improvement.** Inspired by Agarwal et al. [2024], we decompose the regret at any round $t$ by $\sum_i \texttt{Term}_{t,i}$, where $\texttt{Term}_{t,i}$ captures the difference in expected reward when replacing the optimistic distribution of the $i$-th box, denoted by $\hat{\mathcal{E}}_{t,i}$, with its true distribution $D_i$, while keeping all other boxes fixed. Agarwal et al. [2024] bound $\texttt{Term}_{t,i}$ by $\mathcal{O}(\texttt{TV}(\hat{\mathcal{E}}_{t,i}, D_i))$, where TV denotes the total variation distance. However, simply bounding $\texttt{Term}_{t,i}$ by the TV distance results in $\widetilde{\mathcal{O}}(n\sqrt{T})$ regret bound. To obtain a tighter bound, we take a different approach to handle $\texttt{Term}_{t,i}$. Let $\widetilde{R}_i(\sigma_t, z)$ be the expected utility when using threshold vector $\sigma_t$, and the $i$-th box has a fixed value $z$. We then write $\texttt{Term}_{t,i} = \int_0^1 \left( F_{D_i}(z) - F_{\hat{\mathcal{E}}_{t,i}}(z) \right) \frac{\partial}{\partial z} \widetilde{R}_i(\sigma_t; z) \, dz$ where $F_D$ is the cumulative density function of distribution $D$. Our main techniques to bound this term are as follows.

**Technique 1: Bernstein-type Bound.** Both our analysis and algorithm design are based on a Bernstein-type concentration bound. In particular, our algorithm adaptively adjusts the probability mass when constructing empirical distributions, guided by this bound. This introduces an extra factor $\sqrt{F_{D_i}(\cdot)\left(1 - F_{D_i}(\cdot)\right)}$ in the upper bound on $|F_{D_i}(\cdot) - F_{\mathcal{E}_{t,i}}(\cdot)|$, where $\mathcal{E}_{t,i}$ is the *empirical distribution* constructed at round $t$ by assigning equal probability mass to iid samples from $D_i$.

**Technique 2: Sharp Bound of $\frac{\partial}{\partial z} \widetilde{R}_i(\sigma_t, z)$.** We show 1-Lipschitz and monotonically-increasing properties for $\widetilde{R}_i(\sigma_t, z)$, which imply that $0 \leq \frac{\partial}{\partial z} \widetilde{R}_i(\sigma_t, z) \leq 1$. Moreover, when $z$ is small, we prove a sharper bound where $\frac{\partial}{\partial z} \widetilde{R}_i(\sigma_t, z)$ is bounded by the probability that the first $i-1$ boxes have values no larger than $z$.

Next, we apply the techniques above to obtain a sharper bound for $\texttt{Term}_{t,i} = \int_0^1 \left( F_{D_i}(z) - F_{\hat{\mathcal{E}}_{t,i}}(z) \right) \frac{\partial}{\partial z} \widetilde{R}_i(\sigma_t; z) \, dz$. The high-level idea is as follows: when $z$ is large, the Bernstein-type bound for $(F_{D_i}(z) - F_{\hat{\mathcal{E}}_{t,i}}(z))$ contributes to a factor $\sqrt{F_{D_i}(z)\left(1 - F_{D_i}(z)\right)}$, which is small since a large $z$ implies a small $1 - F_{D_i}(z)$; on the other hand, when $z$ is small, we use the sharper bound on $\frac{\partial}{\partial z} \widetilde{R}_i(\sigma_t, z)$ to obtain a tighter bound.

**Techniques for Contextual Linear Case.** We first build a *virtual empirical distribution* $\hat{D}_{t,i}$ for each box $i$ at round $t$, by using all available noise samples from this box plus the mean of the current round. This is virtual since we only observe non-i.i.d. reward samples, not the raw noises. To overcome this issue, we introduce a value-optimistic empirical distribution $\hat{\mathcal{E}}_{t,i}$ by using the linear mean model to shift the observed samples to overestimate $\hat{D}_{t,i}$. We then decompose $\texttt{Term}_{t,i}$ into three terms, two of which can be handled in a similar way as in the non-contextual case. The remaining piece is $\mathbb{E}_{z \sim \mathcal{E}_{t,i}}[\widetilde{R}_i(\sigma_{\hat{\mathcal{E}}_t}; z)] - \mathbb{E}_{z \sim \hat{D}_{t,i}}[\widetilde{R}_i(\sigma_{\hat{\mathcal{E}}_t}; z)]$ which measures the mean estimation error. This term is bounded by applying the 1-Lipschitz property of $\widetilde{R}_i(\sigma_{\hat{\mathcal{E}}_t}; z)$ and contextual linear bandit techniques.

## 2 Preliminaries

In this section, we introduce online Pandora's Box problem in both non-contextual and contextual settings. Due to the limited space, the Prophet Inequality problem is deferred to Appendix A.

**Online Pandora's Box.** In this problem, the learner is given $n$ boxes, each with a known cost $c_i \in [0, 1]$ and an unknown fixed reward distribution $D_i$ supported on $[0, 1]$. The learner plays a $T$-round repeated game where at each round $t \in [T]$, the learner decides an order to inspect boxes

---

**Algorithm 1** Generic descending threshold algorithm for Pandora's Box

---

**Input**: threshold $\sigma = (\sigma_1, \ldots, \sigma_n)$.
**Initialize**: $V_{\max} = -\infty$. Let $\pi_i \in [n]$ be the box with the $i$-th largest threshold.
The environment generates an unknown realization $v = (v_1, \ldots, v_n)$.
**for** $i = 1, \ldots, n$ **do**

    Pay $c_{\pi_i}$ to open box $\pi_i$, observe $v_{\pi_i}$, and update $V_{\max} = \max\{V_{\max}, v_{\pi_i}\}$.
    If $i = n$ or $V_{\max} \geq \sigma_{\pi_{i+1}}$, then stop and take reward $V_{\max}$.

---

and a stopping rule. Upon opening a box $i$, the learner pays cost $c_i$ and receives a reward $v_{t,i} \in [0, 1]$ independently sampled from $D_i$. Once the stopping rule is met, she selects the highest observed reward so far. The utility for the round is the chosen reward minus the total cost of opened boxes.

**Contextual Linear Pandora's Box.** This setting extends the above by allowing each box's distribution to vary across rounds. Specifically, each box $i \in [n]$ is associated with a known cost $c_i \in [0, 1]$, an unknown parameter $\theta_i \in \mathbb{R}^d$, and a fixed unknown noise distribution $D_i$ with zero-mean, supported on $[-\frac{1}{4}, \frac{1}{4}]$. At the beginning of each round $t \in [T]$, the learner observes a context $x_{t,i} \in \mathbb{R}^d$ for each box $i \in [n]$. These contexts $\{x_{t,i}\}_{i \in [n]}$ can be chosen arbitrarily by an adaptive adversary. Based on these contexts and past observations, the learner then decides an inspection order and a stopping rule to open boxes. Different from the non-contextual setup, upon opening box $i$, the learner observes the reward $v_{t,i} = \eta_{t,i} + \mu_{t,i}$ where $\mu_{t,i} = \theta_i^\top x_{t,i} \in [\frac{1}{4}, \frac{3}{4}]$ is the mean and $\eta_{t,i}$ is a noise independently drawn from $D_i$.[2] That is, the reward is drawn from the $[0,1]$-bounded distribution $D_{t,i} = \mu_{t,i} + D_i$ which shifts $D_i$ by the context-dependent mean $\mu_{t,i}$. Following standard assumptions in contextual linear bandits, we assume that $\|\theta_i\|_2 \leq 1$ and $\|x_{t,i}\|_2 \leq 1$ for all $t, i$.

**Optimal Policy and Regret.** To measure the performance, we compete our algorithm against the per-round optimal policy. For both settings, the optimal policy at round $t$ is the Weitzman's algorithm [Weitzman, 1978] which operates with the distribution $D_t = (D_{t,1}, \ldots, D_{t,n})$. In the non-contextual setting, the distribution is fixed across rounds i.e., $D_t = D = (D_1, \ldots, D_n)$, and thus, we will retain the notation $D_t$ in the following. More specifically, Weitzman's algorithm proceeds by computing a threshold $\sigma_{t,i}^*$ for each box $i$, which is the solution of $\mathbb{E}_{x \sim D_{t,i}}[(x - \sigma_{t,i}^*)_+] = c_i$ where $(x)_+ = \max\{0, x\}$. Then, the algorithm inspects the boxes in descending order of $\sigma_{t,i}^*$ and stops as soon as the maximum observed reward so far exceeds the threshold of the next unopened box.

To formally define regret, we introduce the following notations. For any product distribution $D = (D_1, \ldots, D_n)$, let $\sigma_D = (\sigma_{D,1}, \ldots, \sigma_{D,n})$ denote the optimal threshold vector, where each $\sigma_{D,i}$ is the solution of $\mathbb{E}_{x \sim D_i}[(x - \sigma_{D,i})_+] = c_i$. For any threshold vector $\sigma = (\sigma_1, \ldots, \sigma_n) \in [0, 1]^n$ and reward realization $v = (v_1, \ldots, v_n) \in [0, 1]^n$, we use $W(\sigma; v) \subseteq [n]$ to denote boxes opened by Algorithm 1. Further, we define the utility function and expected utility function respectively as:

$$R(\sigma; v) = \max_{i \in W(\sigma; v)} v_i - \sum_{i \in W(\sigma; v)} c_i, \quad \text{and} \quad R(\sigma; D) = \mathbb{E}_{v \sim D}[R(\sigma; v)].$$

With these definitions, the optimal expected utility is $R(\sigma_{D_t}; D_t)$ since when $v \sim D_t$, the set $W(\sigma_{D_t}; v)$ corresponds to the boxes opened by the Weitzman's algorithm. Let $\sigma_t \in \mathbb{R}^n$ denote the threshold vector selected by the algorithm at round $t$. Throughout the paper, all our algorithms follow Algorithm 1 using $\sigma_t$ as input at round $t$, and thus the expected utility our algorithm at round $t$ is $R(\sigma_t; D_t)$. The cumulative regret in $T$ rounds $\text{Reg}_T$ is then defined as follows: $\text{Reg}_T = \mathbb{E}\left[\sum_{t=1}^{T} R(\sigma_{D_t}; D_t) - R(\sigma_t; D_t)\right]$, where $D_t = D$ for the non-contextual case.

# 3 $\widetilde{\mathcal{O}}(\sqrt{nT})$ Regret Bound for Non-Contextual Pandora's Box

In this section, we first present Algorithm 2 for the non-contextual Pandora's Box problem and the main result, and then discuss the analysis.

---

[2]The boundedness assumptions on the noise and mean ensure that rewards lie in $[0, 1]$, which avoids re-deriving standard results under a different scaling. Our results readily extends to sub-Gaussian rewards. We refer readers to Appendix C.7 for details.

## 3.1 Algorithm and Main Result

While Algorithm 2 is inspired by the principle of *optimism in the face of uncertainty*, its application differs from that in the classic multi-armed bandit setting, such as in the Upper Confidence Bound (UCB) algorithm [Auer et al., 2002]. In bandit problems, optimism is typically applied to the mean reward of each arm. In contrast, the Pandora's Box problem requires learning the entire distribution of each box, particularly its cumulative distribution function (CDF). We follow a similar idea to Agarwal et al. [2024], but employ a different approach in the construction of an optimistic estimate of the distribution. The threshold $\sigma_t$ is computed from this optimistic distribution, which implicitly ensures optimism in the algorithm's decisions. More formally, we begin by introducing the notion of stochastic dominance.

**Definition 3.1.1** (**Stochastic dominance**)**.** *Let $D$ and $E$ be two probability distributions with CDFs $F_D$ and $F_E$, respectively. If $\mathbb{P}_{X \sim E}(X \geq a) \geq \mathbb{P}_{Y \sim D}(Y \geq a)$ for all $a \in \mathbb{R}$, we say that $E$ stochastically dominates $D$, and we denote this by $E \succeq_{SD} D$.*

For any two product distributions $D = (D_1, \ldots, D_n)$ and $\mathcal{E} = (\mathcal{E}_1, \ldots, \mathcal{E}_n)$, we use $\mathcal{E} \succeq_{\text{SD}} D$ to indicate that $\mathcal{E}_i \succeq_{\text{SD}} D_i$ for all $i \in [n]$. Let $m_{t,i}$ be the number of samples for each box $i$ at round $t$ and let $t_i(j)$ be the round that the $j$-th reward sample of box $i$ is observed by the learner. We define empirical distribution and optimistic distribution as follows.

- **Empirical distribution $\mathcal{E}_{t,i}$.** For each box $i \in [n]$, we use $m_{t,i}$ i.i.d. samples $\{v_{t_i(j),i}\}_{j=1}^{m_{t,i}}$ to construct $\mathcal{E}_{t,i}$ with respect to the underlying distribution $D_i$ by assigning each sample with probability mass $\frac{1}{m_{t,i}}$. Specifically, $F_{\mathcal{E}_{t,i}}(x) = \frac{1}{m_{t,i}} \sum_{j=1}^{m_{t,i}} \mathbb{I}\{v_{t_i(j)} \leq x\}$ for all $x \in [0, 1]$.

- **Optimistic distribution $\hat{\mathcal{E}}_{t,i}$.** Let $L = 4\log(2nT^2/\delta)$. The CDF of $\hat{\mathcal{E}}_{t,i}$ is constructed as follows:

$$
F_{\hat{\mathcal{E}}_{t,i}}(x) = \begin{cases} 1, & x = 1; \\ \max\left\{0, F_{\mathcal{E}_{t,i}}(x) - \sqrt{\frac{2F_{\mathcal{E}_{t,i}}(x)(1 - F_{\mathcal{E}_{t,i}}(x))L}{m_{t,i}}} - \frac{L}{m_{t,i}}\right\}, & 0 \leq x < 1. \end{cases} \tag{1}
$$

As verified in Lemma C.1.4, the construction in Eq. (1) ensures that $F_{\hat{\mathcal{E}}_{t,i}}(\cdot)$ is a valid CDF. We denote $\mathcal{E}_t = (\mathcal{E}_{t,1}, \ldots, \mathcal{E}_{t,n})$ and $\hat{\mathcal{E}}_t = (\hat{\mathcal{E}}_{t,1}, \ldots, \hat{\mathcal{E}}_{t,n})$. This type of construction has been previously used in [Guo et al., 2019, 2021], but our use differs substantially in the analysis. The following lemma shows that the optimistic product distribution stochastically dominates the underlying product distribution.

**Lemma 3.1.2.** *With probability at least $1 - \delta$, for all $t \in [T]$, we have $\hat{\mathcal{E}}_t \succeq_{SD} D$.*

It is noteworthy that our approach to constructing the optimistic distribution is more adaptive than that of [Agarwal et al., 2024]. In particular, Agarwal et al. [2024] move a fixed amount of probability mass, approximately $1/\sqrt{m_{t,i}}$, to the maximal value that $D_i$ can take. In contrast, we adjust the CDF based on a data-dependent confidence interval, resulting in a more fine-grained and adaptive optimistic distribution. As stated in the following theorem, this construction results in a tighter regret bound.

**Theorem 3.1.3.** *For Pandora's Box problem, with $\delta = T^{-1}$ Algorithm 2 ensures $\text{Reg}_T = \widetilde{\mathcal{O}}(\sqrt{nT})$.*

Again, our bound matches the $\Omega(\sqrt{nT})$ lower bound of Gatmiry et al. [2024] up to logarithmic factors. It is worth noting that their lower bound is proven under the easier full-information setting and thus applicable to our setting.

**Remark 3.1.4** (**High-probability regret bound**)**.** *Under regret metric $\sum_{t=1}^{T} (R(\sigma_D; D) - R(\sigma_t; D))$, our proposed algorithm also achieves a high-probability bound of $\widetilde{O}(\sqrt{nT})$ via two modifications only in our analysis. We refer readers to Appendix C.6 for details.*

## 3.2 Analysis

In this subsection, we sketch the main proof ideas for Theorem 3.1.3 and summarize the new analytical techniques. We start by leveraging a monotonicity property for Pandora's Box problem, established by Guo et al. [2021]: for any distributions $D, E$, if $E \succeq_{SD} D$, then $R(\sigma_E; E) \geq R(\sigma_D; D)$. Since

---

**Algorithm 2** Near-optimal algorithm for Pandora's Box

---

**Input**: confidence $\delta \in (0, 1)$, horizon $T$.
**Initialize**: open all boxes once and observe rewards. Set $m_{1,i} = 1$ for all $i \in [n]$.
**for** $t = 1, 2, \ldots, T$ **do**

> For each $i \in [n]$, construct an optimistic distribution $\hat{\mathcal{E}}_{t,i}$ according to Eq. (1).
> Compute $\sigma_t = (\sigma_{t,1}, \ldots, \sigma_{t,n})$ where $\sigma_{t,i}$ is the solution of $\mathbb{E}_{x \sim \hat{\mathcal{E}}_{t,i}} \left[ (x - \sigma_{t,i})_+ \right] = c_i$.
> Run Algorithm 1 with $\sigma_t$ to open a set of boxes, denoted by $\mathcal{B}_t$, and observe rewards $\{v_{t,i}\}_{i \in \mathcal{B}_t}$.
> Update counters $m_{t+1,i} = m_{t,i} + 1, \forall i \in \mathcal{B}_t$ and $m_{t+1,i} = m_{t,i}, \forall i \notin \mathcal{B}_t$.

---

the algorithm uses threshold $\sigma_t = \sigma_{\hat{\mathcal{E}}_t}$ at each round $t$ and Lemma 3.1.2 shows $\hat{\mathcal{E}}_t \succeq_{\text{SD}} D$, the regret at round $t$, denoted by $\text{Reg}(t)$, is bounded as

$$\text{Reg}(t) := \mathbb{E}\left[ R(\sigma_D; D) - R(\sigma_t; \hat{\mathcal{E}}_t) + R(\sigma_t; \hat{\mathcal{E}}_t) - R(\sigma_t; D) \right] \leq \mathbb{E}\left[ R(\sigma_t; \hat{\mathcal{E}}_t) - R(\sigma_t; D) \right].$$

For simplicity of the exposition, without loss of generality, we assume that $\sigma_{t,1} \geq \sigma_{t,2} \geq \cdots \geq \sigma_{t,n}$. Following Agarwal et al. [2024], we define

$$\mathcal{H}_{t,0} = \hat{\mathcal{E}}_t, \quad \text{and} \quad \mathcal{H}_{t,i} = (D_1, \cdots, D_i, \hat{\mathcal{E}}_{t,i+1}, \cdots, \hat{\mathcal{E}}_{t,n}) \quad \text{for all } t, i. \tag{2}$$

Therefore $\mathcal{H}_{t,n} = D$ so that

$$R(\sigma_t; \hat{\mathcal{E}}_t) - R(\sigma_t; D) = \sum_{i=1}^{n} R(\sigma_t; \mathcal{H}_{t,i-1}) - R(\sigma_t; \mathcal{H}_{t,i}). \tag{3}$$

The decomposition in Eq. (3) breaks the reward difference into a sequence of steps, where each step replaces one coordinate of the optimistic distribution with its true counterpart, thereby allowing for isolation of the effect of each individual coordinate change.

Let $E_{\sigma,i}$ be the event that box $i$ is opened under the threshold $\sigma$, and let $E_{\sigma,i}^c$ be the corresponding complementary event. We define $Q_{t,i}$ as the probability that $\sigma_t$ opens box $i$ given the true product distribution $D$. Using the same argument as in [Agarwal et al., 2024], we have that

$$\mathbb{E}\left[ R(\sigma_t; \mathcal{H}_{t,i-1}) - R(\sigma_t; \mathcal{H}_{t,i}) \right]$$
$$= \mathbb{E}\Big[ Q_{t,i} \underbrace{\left( \mathbb{E}_{x \sim \mathcal{H}_{t,i-1}} \left[ R\left(\sigma_t; x\right) \mid E_{\sigma_t,i} \right] - \mathbb{E}_{y \sim \mathcal{H}_{t,i}} \left[ R\left(\sigma_t; y\right) \mid E_{\sigma_t,i} \right] \right)}_{\text{Term}_{t,i}} \Big]. \tag{4}$$

The key step is to carefully bound $\text{Term}_{t,i}$, and this is where our analysis *fundamentally diverges* from Agarwal et al. [2024]. Before presenting our analysis, we explain below why the previous analysis fails to achieve the $\widetilde{\mathcal{O}}(\sqrt{nT})$ regret bound.

$\widetilde{\mathcal{O}}(Un\sqrt{T})$ **bound of Agarwal et al. [2024].** In the analysis of Agarwal et al. [2024], if the product distribution $D$ is discrete, then they bound $\text{Term}_{t,i} \leq \mathcal{O}(U \cdot \text{TV}(D_i, \hat{\mathcal{E}}_{t,i}))$ (cf. [Agarwal et al., 2024, Lemma 1.3]) where $U$ is the upper bound of the absolute value of function $R$ and could be as large as $n$. Therefore,

$$\text{Reg}_T = \sum_{t=1}^{T} \text{Reg}(t) \leq \sum_{t=1}^{T} \mathbb{E}\left[ R(\sigma_t; \hat{\mathcal{E}}_t) - R(\sigma_t; D) \right] \leq U \sum_{t=1}^{T} \sum_{i=1}^{n} \mathcal{O}\left( \mathbb{E}\left[ Q_{t,i} \text{TV}(D_i, \hat{\mathcal{E}}_{t,i}) \right] \right).$$

Since their algorithm moves probability mass by a fixed amount, approximately $\widetilde{\mathcal{O}}(1/\sqrt{m_{t,i}})$ for each distribution $i$, a simple calculation gives $\text{TV}(D_i, \hat{\mathcal{E}}_{t,i}) \leq \widetilde{\mathcal{O}}(1/\sqrt{m_{t,i}})$, which in turn implies that

$$\text{Reg}_T \leq \widetilde{\mathcal{O}}\left( U \cdot \mathbb{E}\left[ \sum_{t=1}^{T} \sum_{i=1}^{n} \frac{Q_{t,i}}{\sqrt{m_{t,i}}} \right] \right) \leq \widetilde{\mathcal{O}}\left( U \cdot \sqrt{\mathbb{E}\left[ n \sum_{t=1}^{T} \sum_{i=1}^{n} Q_{t,i} \right]} \right) \leq \widetilde{\mathcal{O}}\left( Un\sqrt{T} \right), \tag{5}$$

where the second inequality follows from the standard analysis of optimistic algorithms (e.g., [Slivkins, 2019]), and the last inequality uses the fact that $\sum_{i=1}^{n} Q_{t,i} \leq n$ for all $t$. It is noteworthy that

$\sum_{i=1}^{n} Q_{t,i} = 1$ appears in multi-armed bandit problems since only one arm is played in each round. However, in the problems we consider, there is a chance to open all boxes in one round.

**Refined analysis for $\widetilde{\mathcal{O}}(\sqrt{nT})$ bound.** To get the $\widetilde{\mathcal{O}}(\sqrt{nT})$ regret bound, it suffices to show $\text{Reg}(t) \leq \widetilde{\mathcal{O}}(\sqrt{\sum_i Q_{t,i}/m_{t,i}})$ since

$$\text{Reg}_T \leq \widetilde{\mathcal{O}}\left(\mathbb{E}\left[\sum_{t=1}^{T}\sqrt{\sum_{i\in[n]}\frac{Q_{t,i}}{m_{t,i}}}\right]\right) \leq \widetilde{\mathcal{O}}\left(\mathbb{E}\left[\sqrt{T\sum_{t=1}^{T}\sum_{i\in[n]}\frac{Q_{t,i}}{m_{t,i}}}\right]\right) \leq \widetilde{\mathcal{O}}\left(\sqrt{nT}\right),$$

where the second inequality follows from the Cauchy–Schwarz inequality, and the last inequality results from the bound $\sum_{t,i} Q_{t,i}/m_{t,i} \leq \widetilde{\mathcal{O}}(n)$. For any threshold $\sigma \in [0,1]^n$ and $z \in [0,1]$, define

$$\widetilde{R}_i(\sigma; z) := \mathbb{E}_{x\sim\mathcal{H}_{t,i}}\left[R\left(\sigma; (x_1, \ldots, x_{i-1}, z, x_{i+1}, \ldots, x_n)\right) \mid E_{\sigma,i}\right]. \tag{6}$$

Based on these definitions and the fact that $E_{\sigma_t,i}$ is independent of $i$-th box's reward, we write

$$\text{Term}_{t,i} = \mathbb{E}_{z\sim\hat{\mathcal{E}}_{t,i}}\left[\widetilde{R}_i(\sigma_t; z)\right] - \mathbb{E}_{z\sim D_i}\left[\widetilde{R}_i(\sigma_t; z)\right].$$

The following lemma is a key enabler for our refined analysis of both problems.

**Lemma 3.2.1** (1-**Lipschitzness and monotonicity**). *Consider the Pandora's box problem. For all $t \in [T]$, $i \in [n]$, the function $\widetilde{R}_i(\sigma_t; z)$ is 1-Lipschitz and monotonically-increasing with respect to $z$.*

Since $\widetilde{R}_i(\sigma_t; z)$ is 1-Lipschitz with respect to $z$, the map $\widetilde{R}_i(\sigma_t; z)$ is absolutely continuous on $[0,1]$, which implies that it is differentiable almost everywhere in $[0,1]$. By the fundamental theorem of calculus, for any $x \in [0,1]$, we have that $\widetilde{R}_i(\sigma_t; x) = \widetilde{R}_i(\sigma_t; 0) + \int_0^x \frac{\partial}{\partial z}\widetilde{R}_i(\sigma_t; z) \, dz$. Then it is straightforward to deduce that

$$\text{Term}_{t,i} = \int_0^1 \left(1 - F_{\hat{\mathcal{E}}_{t,i}}(z)\right)\frac{\partial}{\partial z}\widetilde{R}_i(\sigma_t; z) \, dz - \int_0^1 \left(1 - F_{D_i}(z)\right)\frac{\partial}{\partial z}\widetilde{R}_i(\sigma_t; z) \, dz$$

$$= \int_0^1 \left(F_{D_i}(z) - F_{\hat{\mathcal{E}}_{t,i}}(z)\right)\frac{\partial}{\partial z}\widetilde{R}_i(\sigma_t; z) \, dz$$

$$\leq \underbrace{\int_{\sigma_{t,i+1}}^1 \left|F_{D_i}(z) - F_{\hat{\mathcal{E}}_{t,i}}(z)\right| \, dz}_{A_{t,i}} + \underbrace{\int_0^{\sigma_{t,i+1}} \left(F_{D_i}(z) - F_{\hat{\mathcal{E}}_{t,i}}(z)\right)\frac{\partial}{\partial z}\widetilde{R}_i(\sigma_t; z) \, dz}_{B_{t,i}},$$

where the inequality holds since $\widetilde{R}_i(\sigma_t; z)$ is 1-Lipschitz with respect to $z$ and thus its derivative is bounded in $[-1,1]$ almost everywhere. Here, we decompose $\text{Term}_{t,i}$ into two parts $A_{t,i}$ and $B_{t,i}$ to isolate contributions from different value regions, enabling a problem-dependent analysis that yields a sharper regret bound. An advantage of this analysis is to avoid using [Agarwal et al., 2024, Lemma 1.3], which incurs a linear dependence on $U$. Indeed, we have

$$A_{t,i} \leq \int_{\sigma_{t,i+1}}^1 \left|F_{D_i}(z) - F_{\mathcal{E}_{t,i}}(z)\right| \, dz + \int_{\sigma_{t,i+1}}^1 \left|F_{\mathcal{E}_{t,i}}(z) - F_{\hat{\mathcal{E}}_{t,i}}(z)\right| \, dz$$

$$\leq \mathcal{O}\left(\frac{L}{m_{t,i}} + \int_{\sigma_{t,i+1}}^1 \sqrt{\frac{F_{D_i}(z)(1 - F_{D_i}(z))L}{m_{t,i}}} \, dz + \int_{\sigma_{t,i+1}}^1 \sqrt{\frac{F_{\mathcal{E}_{t,i}}(z)(1 - F_{\mathcal{E}_{t,i}}(z))L}{m_{t,i}}} \, dz\right)$$

$$\leq \mathcal{O}\left(\frac{L}{m_{t,i}} + \sqrt{\frac{(1 - F_{D_i}(\sigma_{t,i+1}))L}{m_{t,i}}} + \sqrt{\frac{(1 - F_{\mathcal{E}_{t,i}}(\sigma_{t,i+1}))L}{m_{t,i}}}\right)$$

$$\leq \mathcal{O}\left(\sqrt{\frac{(1 - F_{D_i}(\sigma_{t,i+1}))L}{m_{t,i}}} + \frac{L}{m_{t,i}}\right),$$

where the second inequality bounds $|F_{D_i}(z) - F_{\mathcal{E}_{t,i}}(z)|$ by a Bernstein-type Dvoretzky-Kiefer-Wolfowitz inequality (cf. Lemma C.1.2 in Appendix C.1) and bounds $|F_{\mathcal{E}_{t,i}}(z) - F_{\hat{\mathcal{E}}_{t,i}}(z)|$ by the

construction Eq. (1); the third inequality bounds $1 - F_{D_i}(z) \leq 1 - F_{D_i}(\sigma_{t,i+1})$, and an analogous inequality holds for $1 - F_{\mathcal{E}_{t,i}}(z)$ since $z \geq \sigma_{t,i+1}$; the last inequality replaces $F_{\mathcal{E}_{t,i}}(\sigma_{t,i+1})$ with $F_{D_i}(\sigma_{t,i+1})$ by paying a term of the same order (cf. Eq. (19)).

To handle the term $B_{t,i}$, instead of bounding the derivative by one as before, we introduce the following lemma which more tightly bounds the derivative by the probability that the first $i-1$ boxes have values no larger than $z$, conditioning on opening box $i$.

**Lemma 3.2.2.** *Suppose that* $\sigma_{t,1} \geq \sigma_{t,2} \geq \cdots \geq \sigma_{t,n}$. *Pandora's Box problem satisfies*

$$0 \leq \frac{\partial}{\partial z} \widetilde{R}_i(\sigma_t; z) \leq \prod_{j<i} F_{D_j}(z)/Q_{t,i}, \quad \forall z \in [0, \sigma_{t,i+1}).$$

For shorthand, define $a_i^z = \prod_{j \leq i} F_{D_j}(z)$. By Lemma 3.2.2 and some calculations (deferred to Appendix C), we have that

$$B_{t,i} \leq \int_0^{\sigma_{t,i+1}} \left| F_{D_i}(z) - F_{\hat{\mathcal{E}}_{t,i}}(z) \right| \frac{\sqrt{a_{i-1}^z}\sqrt{a_{i-1}^z}}{Q_{t,i}} \, dz$$

$$\leq \int_0^{\sigma_{t,i+1}} \left( \left| F_{D_i}(z) - F_{\mathcal{E}_{t,i}}(z) \right| + \left| F_{\mathcal{E}_{t,i}}(z) - F_{\hat{\mathcal{E}}_{t,i}}(z) \right| \right) \sqrt{\frac{a_{i-1}^z}{Q_{t,i}}} \, dz$$

$$\leq \mathcal{O}\left( \frac{L}{m_{t,i}} + \frac{1}{\sqrt{Q_{t,i}}} \int_0^1 \sqrt{a_{i-1}^z(1 - F_{D_i}(z))} \sqrt{\frac{L}{m_{t,i}}} \, dz \right),$$

where the second inequality follows from $a_{i-1}^z \leq a_{i-1}^{\sigma_{t,i+1}} \leq a_{i-1}^{\sigma_{t,i}} = Q_{t,i}$ for all $z < \sigma_{t,i+1}$, and the last inequality repeats the same argument used to bound $A_{t,i}$ to bound $|F_{D_i}(z) - F_{\mathcal{E}_{t,i}}(z)|$ and $|F_{\mathcal{E}_{t,i}}(z) - F_{\hat{\mathcal{E}}_{t,i}}(z)|$, respectively.

Plugging the above bounds of $A_{t,i}$ into Eq. (3) and Eq. (4), the regret contribution of $\{A_{t,i}\}_i$ at round $t$ (omitting $L/m_{t,i}$ and some constant) is bounded by

$$\sum_{i \in [n]} Q_{t,i} \sqrt{\frac{(1 - F_{D_i}(\sigma_{t,i+1}))L}{m_{t,i}}} \leq \sqrt{\sum_{i \in [n]} Q_{t,i}(1 - F_{D_i}(\sigma_{t,i+1}))} \sqrt{\sum_{i \in [n]} \frac{LQ_{t,i}}{m_{t,i}}} \leq \sqrt{\sum_{i \in [n]} \frac{LQ_{t,i}}{m_{t,i}}},$$

where the first inequality follows from the Cauchy-Schwarz inequality, and the second inequality follows from $\sum_{i \in [n]} Q_{t,i}(1 - F_{D_i}(\sigma_{t,i+1})) \leq 1$. Indeed, since $Q_{t,i} = a_{i-1}^{\sigma_{t,i}}$ and $\sigma_{t,1} \geq \sigma_{t,2} \ldots \geq \sigma_{t,n}$, we have that $Q_{t,i} \leq a_{i-1}^{\sigma_{t,j+1}}$, which in turn yields the telescoping sum $\sum_{i=1}^n Q_{t,i}\left(1 - F_{D_i}(\sigma_{t,i+1})\right) \leq \sum_{i=1}^n \left( a_{i-1}^{\sigma_{t,j+1}} - a_i^{\sigma_{t,j+1}} \right) \leq 1$.

Similarly, the regret contribution of $\{B_{t,i}\}_i$ at round $t$ (omitting $L/m_{t,i}$ and some constant) is bounded by

$$\sum_{i \in [n]} \int_0^1 \sqrt{a_{i-1}^z - a_i^z} \sqrt{\frac{LQ_{t,i}}{m_{t,i}}} \, dz \leq \int_0^1 \sqrt{\sum_{i \in [n]} (a_{i-1}^z - a_i^z)} \sqrt{\sum_{i \in [n]} \frac{LQ_{t,i}}{m_{t,i}}} \, dz \leq \sqrt{\sum_{i \in [n]} \frac{LQ_{t,i}}{m_{t,i}}},$$

where the first inequality uses the Cauchy-Schwarz inequality and the second inequality uses the telescoping sum and $a_i^z \leq 1$ for all $i, z$. Hence, the cumulative regret is bounded by $\widetilde{\mathcal{O}}(\sqrt{\sum_i Q_{t,i}/m_{t,i}})$. Combining all the results completes the proof of Theorem 3.1.3.

# 4 $\widetilde{\mathcal{O}}(nd\sqrt{T})$ Regret Bound for Contextual Linear Pandora's Box

In this section, we present Algorithm 5 (in Appendix D) for the contextual linear Pandora's Box problem. While our algorithm inherits the main structure of Algorithm 2, it introduces key modifications to ensure optimism, which is critical in the contextual setting. To ensure the optimism, we need to construct an optimistic product distribution which stochastically dominates the underlying product distribution $D_t$ for each round $t$. In the non-contextual case, the optimistic distribution is constructed by using *iid samples* to first construct empirical distribution and then move the probability mass.

However, due to context-dependent shifts, this strategy no longer applies. To address this, we leverage the fact that the shift in each box's distribution is determined by a fixed but unknown vector $\theta_i$.

A natural idea here is to construct empirical distribution by using samples $\{z^t_{t_i(j),i}\}_{j=1}^{m_{t,i}}$ where $z^t_{t_i(j),i} = \eta_{t_i(j),i} + \mu_{t,i}$, since they can be treated as $m_{t,i}$ iid samples drawn from $D_{t,i}$. However, the learner never directly observes the noise sample $\eta_{t_i(j),i}$. To overcome this, we maintain an estimation $\widehat{\theta}_{t,i}$ at each round $t$ by using ridge regression on past observations. Specifically, for each opened box $i \in \mathcal{B}_t$, we make the following updates: $\widehat{\theta}_{t,i} = V_{t,i}^{-1} \sum_{s \le t: i \in \mathcal{B}_s} x_{s,i} v_{s,i}$ where $V_{t,i} = I + \sum_{s \le t: i \in \mathcal{B}_s} x_{s,i} x_{s,i}^\top$. Let $\mathcal{X}$ denote the context space. By the analysis of linear bandits (e.g., [Abbasi-Yadkori et al., 2011]), for each box $i \in [n]$, there exists a *known* function $\beta_i^t : \mathcal{X} \times (0,1) \to \mathbb{R}_{>0}$ such that with probability at least $1 - \delta$, for all $x \in \mathcal{X}$ and all $t$,

$$\left| x^\top (\theta_i - \widehat{\theta}_{t-1,i}) \right| \le \beta_i^t(x, \delta), \text{ where } \beta_i^t(x, \delta) = \mathcal{O}\left( \|x\|_{V_{t-1,i}^{-1}} \sqrt{\log(n\delta^{-1}) + d\log(1 + T/d)} \right).$$

With a confidence radius of $\theta$, we construct a *value-optimistic* empirical distribution $\mathcal{E}_{t,i}$ by assigning each of the following with equal probability mass $m_{t,i}^{-1}$.

$$\hat{z}^t_{t_i(j),i} := \min\left\{ 1, \underbrace{v_{t_i(j),i} - \text{LCB}_i^t(\widehat{\theta}_{t-1,i}, x_{t_i(j),i})}_{\text{Optimistic debiased noise sample}} + \underbrace{\text{UCB}_i^t(\widehat{\theta}_{t-1,i}, x_{t,i})}_{\text{Optimistic estimated mean}} \right\}, \tag{7}$$

where for any context $x \in \mathcal{X}$, any failure probability $\delta \in (0,1)$, and any $\theta \in \mathbb{R}^d$

$$\text{UCB}_i^t(\theta, x) := x^\top \theta + \beta_i^t(x, \delta), \quad \text{and} \quad \text{LCB}_i^t(\theta, x) := x^\top \theta - \beta_i^t(x, \delta). \tag{8}$$

The construction of $\mathcal{E}_{t,i}$ is *value-optimistic* because $\hat{z}^t_{t_i(j),i} \ge z^t_{t_i(j),i}$ holds. If one has $m_{t,i}$ samples at round $t$ for box $i$, then $\mathcal{E}_{t,i}$ stochastically dominates the empirical distribution constructed by using samples $\{z^t_{t_i(j),i}\}_{j=1}^{m_{t,i}}$. Then, for each box $i$, we construct the optimistic distribution $\hat{\mathcal{E}}_{t,i}$ by moving a small amount probability mass for the value-optimistic empirical distribution $\mathcal{E}_{t,i}$ so that it dominates the true distribution (the detailed construction of $\hat{\mathcal{E}}_{t,i}$ and the proof are deferred to Appendix D).

**Lemma 4.1.** *With probability at least $1 - \delta$, for all $t \in [T]$, $\hat{\mathcal{E}}_t \succeq_{SD} D_t$.*

Once the optimism is ensured, we prove the following regret bound for Algorithm 5 in Appendix D.

**Theorem 4.2.** *For contextual linear Pandora's Box problem, choosing $\delta = T^{-1}$ for Algorithm 5 ensures $\text{Reg}_T = \widetilde{\mathcal{O}}(nd\sqrt{T})$ regret bound.*

While the algorithm of [Atsidakou et al., 2024] applies to our setting with $\widetilde{\mathcal{O}}(nT^{5/6})$ regret, Theorem 4.2 gives a $\sqrt{T}$-type regret bound in the contextual linear setting, which is better whenever $d = o(T^{1/3})$. The tightness of $\widetilde{\mathcal{O}}(nd\sqrt{T})$ remains an open question. In the contextual linear bandit setting with $\theta_1 = \cdots = \theta_n$, the known lower bound is $\Omega(d\sqrt{T})$ [Dani et al., 2008]. Our setting however requires learning a separate noise distribution for each box, and we conjecture that this makes the linear dependence on $n$ unavoidable, even when all boxes share the same $\theta$.

**Analysis.** As Lemma 4.1 ensures that $\hat{\mathcal{E}}_t \succeq_{SD} D_t$ for all $t$, we follow the same argument in the non-contextual case to arrive at Eq. (4). The decomposition of the main term in Eq. (4) slightly differs from that of the non-contextual case. We introduce an empirical distribution $\hat{D}_{t,i}$ for the decomposition, which is constructed by assigning equal probability mass $m_{t,i}^{-1}$ for each $\{z^t_{t_i(j),i}\}_{j=1}^{m_{t,i}}$.

Notice that conditioning on history before $t$, $\hat{D}_t$ is a product distribution. Then, conditioning on the history before $t$, for any $i$, the main term is rewritten as

$$\text{Term}_{t,i} = \underbrace{\mathbb{E}_{z \sim \hat{\mathcal{E}}_{t,i}}\left[ \widetilde{R}_i(\sigma_t; z) \right] - \mathbb{E}_{z \sim \mathcal{E}_{t,i}}\left[ \widetilde{R}_i(\sigma_t; z) \right]}_{(\text{I})} + \underbrace{\mathbb{E}_{z \sim \mathcal{E}_{t,i}}\left[ \widetilde{R}_i(\sigma_t; z) \right] - \mathbb{E}_{z \sim \hat{D}_{t,i}}\left[ \widetilde{R}_i(\sigma_t; z) \right]}_{(\text{II})}$$

$$+ \underbrace{\mathbb{E}_{z \sim \hat{D}_{t,i}}\left[ \widetilde{R}_i(\sigma_t; z) \right] - \mathbb{E}_{z \sim D_{t,i}}\left[ \widetilde{R}_i(\sigma_t; z) \right]}_{(\text{III})}.$$

Here, term (I) captures the error of optimistic reweighting, which arises from shifting probability mass in the empirical distribution $\mathcal{E}_{t,i}$ to construct the optimistic distribution $\hat{\mathcal{E}}_{t,i}$. Term (III) quantifies the estimation error between the empirical distribution $\hat{D}_{t,i}$ and the true underlying distribution $D_i$. Both can be bounded in a similar way as in the non-contextual case. It thus remains to handle (II), which captures the error incurred by the optimistic value shifts. One can show that

$$(\text{II}) = \frac{1}{m_{t,i}} \sum_{j=1}^{m_{t,i}} \left( \widetilde{R}(\sigma_t; \hat{z}_{t_i(j),i}) - \widetilde{R}(\sigma_t; z_{t_i(j),i}) \right) \leq \frac{1}{m_{t,i}} \sum_{j=1}^{m_{t,i}} \left| z^t_{t_i(j),i} - \hat{z}^t_{t_i(j),i} \right|,$$

where the inequality uses 1-Lipschitz property (see Lemma 3.2.1). Following standard linear bandit analysis (e.g., [Abbasi-Yadkori et al., 2011]), the total regret incurred by (II) for all boxes is bounded by (omitting constant)

$$\sum_{t=1}^{T} \sum_{i=1}^{n} \mathbb{E} \left[ \frac{Q_{t,i}}{m_{t,i}} \sum_{j=1}^{m_{t,i}} \left| z^t_{t_i(j),i} - \hat{z}^t_{t_i(j),i} \right| \right] \leq \sum_{t=1}^{T} \sum_{i=1}^{n} \mathbb{E} \left[ Q_{t,i} \left( \beta_i^t(x_{t,i}, \delta) + \frac{1}{m_{t,i}} \sum_{j=1}^{m_{t,i}} \beta_i^t(x_{t_i(j),i}, \delta) \right) \right],$$

The first summation term can be bounded by

$$\sum_{t=1}^{T} \sum_{i=1}^{n} \mathbb{E} \left[ Q_{t,i} \beta_i^t(x_{t,i}, \delta) \right] \leq \widetilde{\mathcal{O}} \left( \sum_{i=1}^{n} \mathbb{E} \left[ \sum_{j=1}^{m_{T,i}} \min\{1, \|x_{t_i(j),i}\|_{V^{-1}_{t_i(j)-1,i}}\} \right] \right) \leq \widetilde{\mathcal{O}}(nd\sqrt{T}),$$

where the last inequality applies the Cauchy–Schwarz inequality followed by the elliptical potential lemma [Abbasi-Yadkori et al., 2011, Lemma 11]. The second summation term can be bounded similarly. We refer readers to Appendix D for the omitted details.

## 5  Conclusion and Future Work

In this paper, we propose new algorithms for the Pandora's Box problem in both non-contextual and contextual settings. In the non-contextual case, our algorithm achieves a regret of $\widetilde{\mathcal{O}}(\sqrt{nT})$ which matches the known lower bound up to logarithmic factors. For the contextual case, we design a modified algorithm that attains $\widetilde{\mathcal{O}}(nd\sqrt{T})$ regret. We further extend our methods and analysis to the Prophet Inequality problem in both settings, achieving similar improvements and regret bounds. Our results also elicit compelling open questions:

1. **Minimax regret for Prophet Inequality.** We establish an upper bound of $\widetilde{\mathcal{O}}(\sqrt{nT})$ for Prophet Inequality problem, while Gatmiry et al. [2024] proved a lower bound of $\Omega(\sqrt{T})$, leaving a $\sqrt{n}$-gap. In fact, Jin et al. [2024] show that the optimal sample complexity of Prophet Inequality is $\widetilde{\mathcal{O}}(\frac{1}{\epsilon^2})$, *independent of* $n$, which suggests that a simple explore-then-commit strategy achieves an $n$-independent regret upper bound of $\widetilde{\mathcal{O}}(T^{2/3})$. Whether $\widetilde{\mathcal{O}}(\sqrt{T})$ is achievable remains open.

2. **Tighter regret for contextual linear case.** In the non-contextual case, our bound scales as $\sqrt{n}$, while in the contextual linear setting, it grows linearly with $n$. Can this dependence be improved? It also remains unclear whether the linear dependence on the dimension $d$ is improvable.

## Acknowledgments and Disclosure of Funding

HL is supported by NSF Award IIS-1943607. LJR and JL are supported in part by ONR YIP N000142012571, and NSF awards AF-2312775, CPS-1844729.

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

# Appendix

---

**Algorithm 3** Generic threshold algorithm for Prophet Inequality

---

**Input**: threshold $\sigma = (\sigma_1, \ldots, \sigma_n)$.

The environment generates an unknown realization $v = (v_1, \ldots, v_n)$.

**for** $i = 1, \ldots, n$ **do**
  Open box $i$ to observe $v_i$.
  If $v_i \geq \sigma_i$ or $i = n$, then stop and take reward $v_i$.

---

## A    Problem Setting of Prophet Inequality

**Online Prophet Inequality.** In this problem, the learner is given $n$ boxes, each with an unknown fixed reward distribution $D_i$ supported on $[0, 1]$. The learner plays a $T$-round repeated game where at each round $t \in [T]$, the learner inspects boxes in a fixed order. Upon opening a box $i$, the learner receives a reward $v_{t,i} \in [0, 1]$ independently sampled from $D_i$. Then, she needs to decide immediately whether to accept this reward and stop the process at round $t$ or discard this reward and open the next box. The utility for the round is the chosen reward.

**Contextual Linear Prophet Inequality.** This setting extends the non-contextual one, and the assumptions on reward distributions and boundedness are the same as Section 2.

**Optimal Policy and Regret.** To measure the performance, we compare our algorithm against the per-round optimal policy. Let $D_t = (D_{t,1}, ..., D_{t,n})$ be the product distribution of boxes. The optimal policy for Prophet Inequality in round $t$ is also a *threshold* policy, in which the threshold for each box $i$, denoted by $\sigma_{t,i}^*$ is derived from a *reverse/backward programming*.

**Definition A.1** (**Optimal Threshold on** $D_t$). *Let* $\sigma_{t,n}^* = \mathbb{E}_{X \sim D_{t,n}}[X]$. *For* $i = n - 1, \ldots, 1$, *let* $\sigma_{t,i}^* = \mathbb{E}_{X \sim D_{t,i}}[\max\{X, \sigma_{t,i+1}^*\}]$.

The optimal policy at round $t$ is to run Algorithm 3 with threshold vector $\sigma_t^*$. In fact, the optimal threshold of box $i$ computed from Definition A.1 is equal to the expected reward obtained by running $\sigma_{t,i}^*, \ldots, \sigma_{t,n}^*$ directly from box $i$ [Gatmiry et al., 2024].

More formally, similar to those definitions in Pandora's Box setting, for any product distribution $D = (D_1, ..., D_n)$, let $\sigma_D = (\sigma_{D,1}, \ldots, \sigma_{D,n})$ denote the optimal threshold vector, where each $\sigma_{D,i}$ is the threshold calculated by reverse programming. For any threshold vector $\sigma = (\sigma_1, ..., \sigma_n) \in [0, 1]^n$ and realization $v = (v_1, ..., v_n) \in [0, 1]^n$, we use $S(\sigma; v) \in [n]$ corresponds to the box where Algorithm 3 stops. Further, we define the utility function and expected utility function, respectively:

$$R(\sigma; v) = v_{S(\sigma; v)}, \quad \text{and} \quad R(\sigma; D) = \mathbb{E}_{v \sim D}[R(\sigma; v)]. \tag{9}$$

With these definitions, the optimal expected utility is $R(\sigma_{D_t}; D_t)$ since when $v \sim D_t$, $S(\sigma_{D_t}; v) \in [n]$ corresponds to the box chosen by the optimal policy. Let $\sigma_t \in \mathbb{R}^n$ denote the threshold vector selected by the algorithm at round $t$. Throughout the paper, all our algorithms in the Prophet Inequality setting follow the general structure outlined in Algorithm 3 using $\sigma_t$ as input at round $t$, and thus the expected utility of our algorithms at round $t$ is $R(\sigma_t; D_t)$. The cumulative regret in $T$ rounds $\text{Reg}_T$ is then defined as

$$\text{Reg}_T = \mathbb{E}\left[ \sum_{t=1}^{T} R(\sigma_{D_t}; D_t) - R(\sigma_t; D_t) \right],$$

where $D_t = D$ for the non-contextual case.

## B    Related Work

The Pandora's Box problem, introduced by Weitzman [1978], has been extended in various directions, including matroid constraints [Singla, 2018], correlated distributions [Chawla et al., 2020, Gergatsouli and Tzamos, 2023], order constraints [Boodaghians et al., 2020], and settings without inspection [Fu et al., 2023, Beyhaghi and Cai, 2023]. The Prophet Inequality problem was first studied by Krengel and Sucheston [1977], Samuel-Cahn [1984], with subsequent generalizations to $k$-selection [Alaei, 2014, Jiang et al., 2022], matroid constraints [Kleinberg and Weinberg, 2012, 2019, Feldman et al., 2021], and online matching [Gravin and Wang, 2019, Ezra et al., 2022]. These works typically assume known distributions and focus on achieving constant-factor approximation guarantees.

A growing body of work has begun to study online variants of the Prophet Inequality and Pandora's Box problems [Guo et al., 2019, Fu and Lin, 2020, Guo et al., 2021, Gatmiry et al., 2024, Agarwal et al., 2024]. In the PAC setting, Fu and Lin [2020], Guo et al. [2021] establish a sample complexity upper bound of $\widetilde{\mathcal{O}}(\frac{n}{\epsilon^2})$ for Pandora's Box, matching the known lower bound up to logarithmic factors, where $\epsilon$ denotes the desired accuracy. Guo et al. [2021] further develop a general framework for learning over product distributions, which applies to both problems and yields a sample complexity of $\widetilde{\mathcal{O}}(\frac{n}{\epsilon^2})$ for Prophet Inequality as well. Later, Jin et al. [2024] provide a near-optimal sample complexity $\widetilde{\mathcal{O}}(\frac{1}{\epsilon^2})$ for Prophet Inequality, which removes the linear dependence on $n$. For the regret minimization problem, Gatmiry et al. [2024] establish $\tilde{\mathcal{O}}(n^3\sqrt{T})$ regret bound for Prophet Inequality and $\tilde{\mathcal{O}}(n^{4.5}\sqrt{T})$ regret bound for Pandora's Box under a constrained feedback model, where the learner observes only the final reward without knowing which box produced it. Subsequently, Agarwal et al. [2024] introduce a general framework for monotone stochastic optimization under semi-bandit feedback, achieving $\tilde{\mathcal{O}}(n\sqrt{T})$ regret for both problems. All these works assume a fixed but unknown product distribution. Moreover, despite this progress, the best known regret bounds for both Prophet Inequality and Pandora's Box under semi-bandit feedback remain at $\tilde{\mathcal{O}}(n\sqrt{T})$, which falls short of the known lower bounds [Gatmiry et al., 2024].

To better capture the dynamics in decision-making scenarios, Gergatsouli and Tzamos [2022] study the Pandora's Box problem in the adversarial setting where the reward of each box is chosen by an adversary and obtain sublinear regret bounds by comparing the algorithm against a weak benchmark. Later, Gatmiry et al. [2024] show that in the same setting, when the algorithm competes against the optimal policy, no algorithm can achieve sublinear regret, even with full information feedback. To make the problem tractable while still allowing distributions to change over time, Atsidakou et al. [2024] explore a contextual Pandora's Box model where in each round, the learner observes a context vector for each box, whose optimal threshold can be approximated by a function of the context and an unknown vector. They provide a reduction from contextual Pandora's Box to an instance of online regression. Their algorithm suffers a $\widetilde{\mathcal{O}}(T^{5/6})$ regret bound if the threshold can be linearly approximated. We note that their algorithm for the linear case also applies to our setting (see the discussion below), but our algorithm enjoys $\widetilde{\mathcal{O}}(nd\sqrt{T})$, which is improves on the aforementioned bound whenever $d = o(T^{1/3})$.

**Discussion of [Atsidakou et al., 2024].** Here, we show how the algorithm proposed by Atsidakou et al. [2024] can be applied to our setting. Their work makes the following realizability assumption.

**Assumption 1** (Realizability of [Atsidakou et al., 2024]). *There exists $w_1, \ldots, w_n \in \mathbb{R}^d$ and a function $h$ such that every time $t \in [T]$ and every box $i \in [n]$, the optimal threshold for distribution $D_{t,i}$ is equal to $h(w_i, x'_{t,i})$, i.e., $\sigma^*_{t,i} = h(w_i, x'_{t,i})$ where $x'_{t,i}$ is a context vector of box $i$ at round $t$ and*

$$\mathbb{E}_{X \sim D_{t,i}}\left[(X - \sigma^*_{t,i})_+\right] = c_i.$$

Below, we show that in our setting, the optimal threshold can be realized by a linear function $h$ (that is, $h(a, b) = a^\top b$). To see this, first note that the optimal threshold $\sigma^*_{t,i}$ in our setting satisfies $\mathbb{E}_{X \sim D_{t,i}}\left[(X - \sigma^*_{t,i})_+\right] = c_i$, which, based on the definition of $D_{t,i}$, is equivalent to $\mathbb{E}_{X \sim D_i}\left[(X - (\sigma^*_{t,i} - \mu_{t,i}))_+\right] = c_i$ where $\mu_{t,i} = \theta_i^\top x_{t,i}$. Therefore, if we let $\sigma^*_i$ be the unique solution of $\mathbb{E}_{X \sim D_i}\left[(X - \sigma^*_i)_+\right] = c_i$, then we have

$$\sigma^*_{t,i} = \mu_{t,i} + \sigma^*_i = \langle w_i, x'_{t,i} \rangle = h(w_i, x'_{t,i}),$$

for $w_i = (\theta_{i,1}, \ldots, \theta_{i,d}, \sigma_1, \ldots, \sigma_n) \in \mathbb{R}^{d+n}$, and $x'_{t,i} = ([x_{t,i}]_1, \ldots, [x_{t,i}]_d, 0, \ldots, 1, \ldots, 0) \in \mathbb{R}^{d+n}$, where $x'_{t,i}$ takes value one at the $d + i$ index. This proves that the optimal threshold is realized by a linear function of some context, making the algorithm of [Atsidakou et al., 2024] applicable to our setting.

A seemingly related strand is cascading bandits [Kveton et al., 2015], where the learner presents an ordered list of items to a user, with items corresponding to boxes in our setting. Then, the user scans the list and clicks the first attractive item if any. The learner observes binary click signals for items up to that click, which yields order dependent semi bandit feedback. Despite this shared feedback structure, Pandora's Box and Prophet Inequality differ in who controls how many items are inspected. In cascading bandits the number is user driven since observation stops at the first click. In Pandora's

Box and in Prophet Inequality the number is policy driven since the algorithm adaptively opens boxes and decides when to stop.

# C Omitted Details from the Non-Contextual Settings

## C.1 Technical Lemmas

**Lemma C.1.1** (Bernstein inequality). *Given $m \in \mathbb{N}$, let $X_1, X_2, \ldots, X_m$ be i.i.d. random variables such that $\mathbb{E}[X_i] = 0, \mathbb{E}[X_i^2] = \sigma^2$, and $|X_i| \leq M$ for some constant $M > 0$. Then, for all $t > 0$*

$$\mathbb{P}\Big(\Big|\sum_{i=1}^{m} X_i\Big| > t\Big) \leq 2\exp\Big(-\frac{t^2}{2m\sigma^2 + \frac{2}{3}Mt}\Big).$$

**Lemma C.1.2.** *Let $D = (D_1, \ldots, D_n)$ be a $[0,1]$-bounded product distribution, and for each $i$, let $\mathcal{E}_{N_i,i}$ be the empirical distribution constructed by assigning $N_i$ i.i.d. samples from $D_i$ with equal probability mass $1/N_i$. With probability at least $1 - \delta$, for any $i \in [n]$, any $N_i \in [T]$, and any $z \in [0,1]$,*

$$\left|F_{D_i}(z) - F_{\mathcal{E}_{N_i,i}}(z)\right| \leq \sqrt{\frac{2F_{D_i}(z)(1 - F_{D_i}(z))\log(2T^2n\delta^{-1})}{N_i}} + \frac{\log(2T^2n\delta^{-1})}{N_i}. \tag{10}$$

*Proof.* This proof follows a similar idea to Guo et al. [2019, Lemma 5], but we apply an additional union bound to account for all possible $N_i$. Fix $i$ and $N_i$. It is suffices to show that for all $z \in [0,1]$ such that $F_{D_i}(z)$ is multiples of $1/N_i$, the following holds.

$$\left|F_{D_i}(z) - F_{\mathcal{E}_{N_i,i}}(z)\right| \leq \sqrt{\frac{2F_{D_i}(z)(1 - F_{D_i}(z))\log(2T^2n\delta^{-1})}{N_i}} + \frac{2\log(2T^2n\delta^{-1})}{3N_i}, \tag{11}$$

This is because given the results above, the general bound on the difference of CDF of any value differs at most an extra additive factor $\frac{1}{N_i} \leq \frac{\log(2T^2n\delta^{-1})}{3N_i}$.

Then fix any $z$ such that $F_{D_i}(z)$ is multiples of $1/N_i$. Let $Y_i$ be the indicator function of the $i$-th sample no smaller than $z$ and let $X_i = Y_i - \mathbb{E}[Y_i]$. By Bernstein inequality (see Lemma C.1.1), taking $t$ be $N_i$ times the RHS of Eq. (11),

$$\mathbb{P}\left(\left|\sum_{i=1}^{N_i} X_i\right| > t\right) = \exp\left(-\frac{t^2}{2N_iF_{D_i}(z)(1 - F_{D_i}(z)) + \frac{2}{3}t}\right) \leq \frac{\delta}{nT^2}$$

Since $N_i \leq T$, by a union bound over all $N_i$, with probability at least $1 - \frac{\delta}{nT}$, Eq. (11) holds for all $z$ such that $F_{D_i}(z)$ is multiples of $1/N_i$. Further, by union bounds over all $i \in [n]$ and $N_i \in [T]$, we get that Eq. (10) does not hold with probability at most $\delta$. □

As $F_{D_i}(\cdot)$ is unknown, we present the following corollary, which provides a confidence bound depending on known $F_{\mathcal{E}_i}(\cdot)$.

**Corollary C.1.3.** *Under the same setting of Lemma C.1.2, suppose that the concentration bounds of Lemma C.1.2 all hold, and then for any $i \in [n]$, any $N_i \in [T]$ and any $x \in [0,1]$,*

$$\left|F_{D_i}(x) - F_{\mathcal{E}_{N_i,i}}(x)\right| \leq \sqrt{\frac{2F_{\mathcal{E}_{N_i,i}}(x)(1 - F_{\mathcal{E}_{N_i,i}}(x))L}{N_i}} + \frac{L}{N_i}, \text{ where } L = 4\log(2nT^2\delta^{-1}). \tag{12}$$

*Proof.* Fix any $i$ and $N_i$. For notional simplicity, we write $\mathcal{E}_i = \mathcal{E}_{N_i,i}$. For any $x \in [0,1]$,

$$F_{D_i}(x)(1 - F_{D_i}(x)) \leq F_{\mathcal{E}_i}(x)(1 - F_{\mathcal{E}_i}(x)) + |F_{D_i}(x)(1 - F_{D_i}(x)) - F_{\mathcal{E}_i}(x)(1 - F_{\mathcal{E}_i}(x))|$$
$$\leq F_{\mathcal{E}_i}(x)(1 - F_{\mathcal{E}_i}(x)) + |F_{D_i}(x) - F_{\mathcal{E}_i}(x)|, \tag{13}$$

where the last inequality uses the fact that function $y - y^2$ is 1-Lipschitz for $y \in [0,1]$.

Let $L' = \log(2nT^2\delta^{-1})$. From Eq. (10), we can show for any $x \in [0,1]$

$$|F_{D_i}(x) - F_{\mathcal{E}_i}(x)| \leq \sqrt{\frac{2F_{D_i}(x)(1 - F_{D_i}(x))L'}{N_i}} + \frac{L'}{N_i}$$

$$\leq \sqrt{\frac{2L'\left(F_{\mathcal{E}_i}(x)(1 - F_{\mathcal{E}_i}(x)) + |F_{D_i}(x) - F_{\mathcal{E}_i}(x)|\right)}{N_i}} + \frac{L'}{N_i}$$

$$\leq \sqrt{\frac{2L'F_{\mathcal{E}_i}(x)(1 - F_{\mathcal{E}_i}(x))}{N_i}} + \sqrt{\frac{2L'|F_{D_i}(x) - F_{\mathcal{E}_i}(x)|}{N_i}} + \frac{L'}{N_i}$$

$$\leq \sqrt{\frac{2L'F_{\mathcal{E}_i}(x)(1 - F_{\mathcal{E}_i}(x))}{N_i}} + \frac{|F_{D_i}(x) - F_{\mathcal{E}_i}(x)|}{2} + \frac{2L'}{N_i}, \qquad (14)$$

where the second inequality uses Eq. (13), and the last inequality uses $2\sqrt{ab} \leq a + b$ for any $a, b \geq 0$ (here $a = L'/N_i$ and $b = \frac{1}{2}|F_{D_i}(x) - F_{\mathcal{E}_i}(x)|$). Rearranging the above inequality gives the claimed bound for fixed $i, N_i$. Since this argument holds for all $i, N_i$, the lemma thus follows. $\qquad\square$

**Lemma C.1.4.** *For all $t, i$, $F_{\hat{\mathcal{E}}_{t,i}}(\cdot)$ constructed by Eq. (1) is a valid CDF.*

*Proof.* To show $F_{\hat{\mathcal{E}}_{t,i}}(\cdot)$ is valid, we show that it is a right-continuous, non-decreasing function in the range of $[0,1]$. Consider any given $t, i$. From Eq. (1), one can easily see $F_{\hat{\mathcal{E}}_{t,i}}(x) \in [0,1]$ for all $x \in [0,1]$. As $F_{\mathcal{E}_{t,i}}(\cdot)$ is right-continuous and max function preserves the right-continuous property, $F_{\hat{\mathcal{E}}_{t,i}}(\cdot)$ is again right-continuous.

Now, we verify the monotonicity of $F_{\hat{\mathcal{E}}_{t,i}}(\cdot)$. Since $F_{\mathcal{E}_{t,i}}(\cdot)$ is non-decreasing, we only need to show $F_{\hat{\mathcal{E}}_{t,i}}(x)$ is non-decreasing as a function of $F_{\mathcal{E}_{t,i}}(x)$. Let $k = \frac{L}{m_{t,i}}$. For the interval of $x$ such that $F_{\mathcal{E}_{t,i}}(x) < k$, $F_{\hat{\mathcal{E}}_{t,i}}(x)$ is 0, and thus it is non-decreasing. Then we only need to consider the interval of $x$ such that $F_{\mathcal{E}_{t,i}}(x) \geq k$. We let $y = F_{\mathcal{E}_{t,i}}(x)$, and rewrite the formulation for $F_{\hat{\mathcal{E}}_{t,i}}(x)$ as $f(y) = y - \sqrt{2ky(1-y)} - k$. In this case, $f'(y) = 1 - \frac{\sqrt{k}(1-2y)}{\sqrt{2y(1-y)}}$. When $y \geq \frac{1}{2}$, this is obviously non-negative, and thus $f(y)$ is non-decreasing. Then, we consider the case of $y < \frac{1}{2}$ and show that $\sqrt{2y(1-y)} > \sqrt{k}(1-2y)$. As $y \geq k$, it is enough to show $\sqrt{2y(1-y)} > \sqrt{y}(1-2y)$. This is equivalent to show $4y^2 - 2y - 1 < 0$ when $y < \frac{1}{2}$, which can be easily verified.

As the argument holds for all $t, i$, the lemma thus follows. $\qquad\square$

**Lemma C.1.5** (Restatement of Lemma 3.1.2)**.** *With probability at least $1 - \delta$, for all $t \in [T]$, $\hat{\mathcal{E}}_t \succeq_{SD} D$.*

*Proof.* For this proof, we show $\hat{\mathcal{E}}_{t,i} \succeq_{SD} D_i$ for all $t, i$. Fix $t, i$. It suffices to show that for all $x \in [0,1]$, $F_{\hat{\mathcal{E}}_{t,i}}(x) \leq F_{D_i}(x)$. For $x = 1$, $F_{\hat{\mathcal{E}}_{t,i}}(x) = F_{D_i}(x) = 1$. For $x \in [0,1)$, we have

$$F_{\hat{\mathcal{E}}_{t,i}}(x) = \max\left\{0, F_{\mathcal{E}_{t,i}}(x) - \sqrt{\frac{2F_{\mathcal{E}_{t,i}}(x)(1 - F_{\mathcal{E}_{t,i}}(x))L}{m_{t,i}}} - \frac{L}{m_{t,i}}\right\}$$

$$\leq \max\left\{0, F_{D_i}(x)\right\} = F_{D_i}(x),$$

where the inequality follows from Corollary C.1.3.

Repeating the same argument for all $i \in [n]$ completes the proof. $\qquad\square$

## C.2 Lipschitz Property, Monotonicity, and Sharp Derivative for Pandora's Box

Before proving Lipschitz property and monotonicity for $\widetilde{R}_i(\sigma_t, z)$, we first introduce the following supporting results.

**Definition C.2.1** ($(n, D, c)$-**Pandora's box instance**). *We use $(n, D, c)$ to characterize a Pandora's box instance where product distribution $D = (D_1, \ldots, D_n)$ is over $n$ boxes, each of which is associated with a cost $c_i$.*

For any $(n, D, c)$-Pandora's box instance, let $W_u : [0, 1]^n \times [0, 1]^n \to 2^{[n]}$ be a general threshold rule with initial value $u$. Specifically, for any initial value $u$, threshold $\sigma$, and realization $x \sim D$, $W_u(\sigma, x)$ is the set of opened boxes following rule $W_u$. For any $u, \sigma$, if the initial value $u \geq \max_i \sigma_i$, then $W_u(\sigma, \cdot) = \emptyset$, that is, the learner does not open any box and keeps the initial value $u$. Otherwise, the learner keeps the initial value $u$ in hand but continues to open boxes by running Algorithm 1 with $V_{\max} = u$ and threshold $\sigma$. Let

$$R(\sigma; x; u; W_v) = \max\left\{u, \max_{i \in W_v(\sigma; x)} x_i\right\} - \sum_{i \in W_v(\sigma; x)} c_i$$

$$R(\sigma; D; u; W_v) = \mathop{\mathbb{E}}_{x \sim D}[R(\sigma; x; u; W_v)].$$

Here, $R(\sigma; x; u; W_v)$ is the utility if the learner adopts rule $W_v$ to run threshold $\sigma$ with initial value $u$ over realization $x$. As shown by Weitzman [1978], for any initial value $u$, the expected utility $R(\sigma_D; D; u; W_u)$ enjoys the optimality.

**Lemma C.2.2** (**Monotonicity and $1$-Lipschitz Property for Pandora's Box**). *For any initial values $u, v$, if the learner runs general threshold rules $W_u$ and $W_v$ with $u \geq v$, respectively on a $(n, D, c)$-Pandora's Box instance, then we have*

$$0 \leq \mathop{\mathbb{E}}_{x \sim D}[R(\sigma_D; x; u; W_u)] - \mathop{\mathbb{E}}_{x \sim D}[R(\sigma_D; x; v; W_v)] \leq u - v.$$

*Consequenctly, the following $1$-Lipschitz property holds: for any $u, v$,*

$$\left|\mathop{\mathbb{E}}_{x \sim D}[R(\sigma_D; x; u; W_u)] - \mathop{\mathbb{E}}_{x \sim D}[R(\sigma_D; x; v; W_v)]\right| \leq |u - v|.$$

*Proof.* Consider any $u \geq v$. For any realization $x = (x_1, \ldots, x_n)$ drawn from product distribution $D$, the utility given the initial value $s$ ($= u$ or $v$) and rule $W$ is

$$R(\sigma_D; x; s; W) = \max\left\{s, \max_{i \in W(\sigma_D; x)} x_i\right\} - \sum_{i \in W(\sigma_D; x)} c_i.$$

Taking $W = W_v$, and $s = u$ and $v$, respectively, we have

$$\mathop{\mathbb{E}}_{x \sim D}[R(\sigma_D; x; u; W_v)] - \mathop{\mathbb{E}}_{x \sim D}[R(\sigma_D; x; v; W_v)]$$
$$= \mathop{\mathbb{E}}_{x \sim D}\left[\max\left\{u, \max_{i \in W_u(\sigma_D; x)} x_i\right\} - \max\left\{v, \max_{i \in W_u(\sigma_D; x)} x_i\right\}\right]$$
$$\geq 0.$$

Given the initial value $u$ and threshold $\sigma_D$, using rule $W_u$ recovers the Weitzman's optimal algorithm. Thus, the corresponding expected utility should be no smaller than that of running rule $W_v$ with the same threshold $\sigma_D$ and initial value $u$, i.e., $\mathbb{E}_{x \sim D}[R(\sigma_D; x; u; W_v)] \leq \mathbb{E}_{x \sim D}[R(\sigma_D; x; u; W_u)]$, which implies $0 \leq \mathbb{E}_{x \sim D}[R(\sigma_D; x; u; W_u)] - \mathbb{E}_{x \sim D}[R(\sigma_D; x; v; W_v)]$.

Taking $W = W_u$, and $s = u$ and $v$, respectively, we have

$$\mathop{\mathbb{E}}_{x \sim D}[R(\sigma_D; x; u; W_u)] - \mathop{\mathbb{E}}_{x \sim D}[R(\sigma_D; x; v; W_u)]$$
$$= \mathop{\mathbb{E}}_{x \sim D}\left[\max\left\{u, \max_{i \in W_u(\sigma_D; x)} x_i\right\} - \max\left\{v, \max_{i \in W_u(\sigma_D; x)} x_i\right\}\right]$$
$$\leq u - v.$$

where the inequality holds since for any $x$, $f_x(v) = \max\{v, x\}$ is 1-Lipschitz with respect to $v$.

As $W_v$ is optimal given initial value $v$ and threshold $\sigma_D$, we have $\mathbb{E}_{x \sim D}[R(\sigma_D; x; v; W_u)] \leq \mathbb{E}_{x \sim D}[R(\sigma_D; x; v; W_v)]$, implying $\mathbb{E}_{x \sim D}[R(\sigma_D; x; u; W_u)] - \mathbb{E}_{x \sim D}[R(\sigma_D; x; v; W_v)] \leq u - v$. Thus, the lemma follows. $\qquad\square$

For any $x \in [0,1]^n$, we define $x_{>i} := (x_{i+1}, \ldots, x_n)$ and $x_{<i} := (x_1, \ldots, x_{i-1})$. Based on the definition of $\mathcal{H}_{t,i} = (D_1, \cdots, D_i, \hat{\mathcal{E}}_{t,i+1}, \cdots, \hat{\mathcal{E}}_{t,n})$, we define

$$\mathcal{H}_{t,>i} := (\hat{\mathcal{E}}_{t,i+1}, \ldots, \hat{\mathcal{E}}_{t,n}) \quad \text{and} \quad \mathcal{H}_{t,<i} := (D_1, \ldots, D_{i-1}).$$

For any $z \in [0,1]$, any $x \in [0,1]^n$, and any $\sigma \in [0,1]^n$, we define

$$\widetilde{R}_i(\sigma; x_{<i}, z) := \mathop{\mathbb{E}}_{x_{>i} \sim \mathcal{H}_{t,>i}} \left[ R\left( \sigma; (x_1, \ldots, x_{i-1}, z, x_{i+1}, \ldots, x_n) \right) \right] \tag{15}$$

$$= \mathop{\mathbb{E}}_{x_{>i} \sim (\hat{\mathcal{E}}_{t,i+1}, \ldots, \hat{\mathcal{E}}_{t,n})} \left[ R\left( \sigma; (x_1, \ldots, x_{i-1}, z, x_{i+1}, \ldots, x_n) \right) \right],$$

where the expectation is taken only for $x_{>i} := (x_{i+1}, \ldots, x_n)$.

**Lemma C.2.3** (Restatement of Lemma 3.2.1)**.** *For all $t \in [T]$ and $i \in [n]$, the map $\widetilde{R}_i(\sigma_t; z)$ is 1-Lipschitz and monotonically-increasing with respect to $z$ in the Pandora's Box problem.*

*Proof.* For simplicity, we assume without loss of generality that $\sigma_{t,1} \geq \sigma_{t,2} \geq \cdots \geq \sigma_{t,n}$. For any $z \in [0,1]$ and any $x \in [0,1]^n$, if box $i$ is opened at round $t$, then the map $\widetilde{R}_i(\sigma_t; x_{<i}, z)$ is the expected utility of running $\sigma_t$ on $(x_{i+1}, \ldots, x_n) \sim \mathcal{H}_{t,>i}$ with initial value $\max\{x_1, \ldots, x_{i-1}, z\}$. A key fact here is that $\sigma_t = \sigma_{\hat{\mathcal{E}}_t}$ and $\mathcal{H}_{t,>i} = (\hat{\mathcal{E}}_{t,i+1}, \ldots, \hat{\mathcal{E}}_{t,n})$, which imply that the algorithm runs the optimal threshold in descending order. Thus Lemma C.2.2 implies that $\widetilde{R}_i(\sigma_t; x_{<i}, z)$ is 1-Lipschitz with respect to $z$ and for any $u \geq v$, and that $\widetilde{R}_i(\sigma_t; x_{<i}, u) - \widetilde{R}_i(\sigma_t; x_{<i}, v) \in [0, u-v]$. This, in turn, implies that

$$\widetilde{R}_i(\sigma_t; u) - \widetilde{R}_i(\sigma_t; v) = \mathop{\mathbb{E}}_{x_{<i} \sim (D_1, \cdots, D_{i-1})} \left[ \widetilde{R}_i(\sigma_t; x_{<i}, u) - \widetilde{R}_i(\sigma_t; x_{<i}, v) \mid E_{\sigma_t, i} \right] \in [0, u-v].$$

The 1-Lipschitz property of $\widetilde{R}_i(\sigma; z)$ is an immediate corollary, and the proof is thus complete. $\square$

**Lemma C.2.4** (Restatement of Lemma 3.2.2)**.** *Suppose that $\sigma_{t,1} \geq \sigma_{t,2} \geq \cdots \geq \sigma_{t,n}$. The Pandora's Box problem satisfies the following:*

$$0 \leq \frac{\partial}{\partial z} \widetilde{R}_i(\sigma_t; z) \leq \frac{\prod_{j<i} F_{D_j}(z)}{Q_{t,i}}, \quad \forall z \in [0, \sigma_{t,i+1}).$$

*Proof.* According to Lemma 3.2.1, the map $\widetilde{R}_i(\sigma_t; z)$ is monotonically increasing with respect to $z$, which implies that $0 \leq \frac{\partial}{\partial z} \widetilde{R}_i(\sigma_t; z)$. Then, we only need to prove for any $0 \leq v \leq u < \sigma_{t,i+1}$,

$$\widetilde{R}_i(\sigma_t; u) - \widetilde{R}_i(\sigma_t; v) \leq (u-v) \cdot \frac{\prod_{j<i} F_{D_j}(u)}{Q_{t,i}}. \tag{16}$$

To this end, we deduce the following:

$$\widetilde{R}_i(\sigma_t; u) - \widetilde{R}_i(\sigma_t; v)$$

$$= \mathop{\mathbb{E}}_{x_{<i} \sim (D_1, \cdots, D_{i-1})} \left[ \widetilde{R}_i(\sigma_t; x_{<i}, u) - \widetilde{R}_i(\sigma_t; x_{<i}, v) \mid E_{\sigma_t, i} \right]$$

$$\overset{(a)}{=} \mathop{\mathbb{E}}_{x_{<i} \sim (D_1, \cdots, D_{i-1})} \left[ \widetilde{R}_i(\sigma_t; x_{<i}, \max\{x_{<i}, u\}) - \widetilde{R}_i(\sigma_t; x_{<i}, \max\{x_{<i}, v\}) \mid E_{\sigma_t, i} \right]$$

$$\leq \mathop{\mathbb{E}}_{x_{<i} \sim (D_1, \cdots, D_{i-1})} \left[ \max\{x_{<i}, u\} - \max\{x_{<i}, v\} \mid E_{\sigma_t, i} \right]$$

$$\overset{(b)}{=} \mathop{\mathbb{E}}_{x_{<i} \sim (D_1, \cdots, D_{i-1})} \left[ \left( \max\{x_{<i}, u\} - \max\{x_{<i}, v\} \right) \cdot \mathbb{I}\left\{ u > \max_{j<i} x_j \right\} \mid E_{\sigma_t, i} \right]$$

$$\leq (u-v) \mathop{\mathbb{E}}_{x_{<i} \sim (D_1, \cdots, D_{i-1})} \left[ \mathbb{I}\left\{ u > \max_{j<i} x_j \right\} \mid E_{\sigma_t, i} \right]$$

$$= (u-v) \cdot \mathop{\mathbb{P}}_{x_{<i} \sim (D_1, \cdots, D_{i-1})} \left( \max_{j<i} x_j < u \mid E_{\sigma_t, i} \right)$$

$$= (u - v) \cdot \frac{\mathbb{P}_{x_{<i} \sim (D_1, \cdots, D_{i-1})} \left( \max_{j<i} x_j < u, E_{\sigma_t, i} \right)}{\mathbb{P}_{x_{<i} \sim (D_1, \cdots, D_{i-1})} \left( E_{\sigma_t, i} \right)}$$

$$\overset{(c)}{=} (u - v) \cdot \frac{\mathbb{P}_{x_{<i} \sim (D_1, \cdots, D_{i-1})} \left( \max_{j<i} x_j < u \right)}{\mathbb{P}_{x_{<i} \sim (D_1, \cdots, D_{i-1})} \left( E_{\sigma_t, i} \right)}$$

$$= (u - v) \frac{\prod_{j<i} F_{D_j}(u)}{Q_{t,i}},$$

where $(a)$ holds since conditioning on $E_{\sigma_t, i}$ (i.e., opening box $i$), the expected reward remains unchanged by replacing $u$ (or $v$) by the maximum so far, the inequality uses the 1-Lipschitz property of $\widetilde{R}_i(\sigma_t; x_{<i}, z)$ with respect to variable $z$, (b) follows from the fact that $\max(x_{<i}, u) - \max(x_{<i}, v)$ is non-zero only when $\max_{j<i} x_j < u$ as $u \geq v$, and (c) holds due to $\{\max_{j<i} x_j < u\} \subseteq E_{\sigma_t, i}$ where this containment follows from the fact that $u < \sigma_{t,i+1} \leq \sigma_{t,i}$ and $E_{\sigma_t, i} = \{\max_{j<i} x_j < \sigma_{t,i}\}$.

Thus, the stated claim holds. $\qquad \square$

### C.3 Lipschitz Property, Monotonicity, and Sharp Derivative for Prophet Inequality

In the following, we prove the similar results for Prophet Inequality.

**Lemma C.3.1.** *Consider any two product distributions $D_u = (u, D_2, D_3, \ldots, D_n)$ and $D_v = (v, D_2, D_3, \ldots, D_n)$, where the first box deterministically produces rewards $u$ and $v$, respectively, and the remaining boxes share the same distributions. If $u \geq v$, then*

$$0 \leq R(\sigma_{D_u}; D_u) - R(\sigma_{D_v}; D_v) \leq u - v.$$

*As a corollary, we have the following 1-Lipschitz property. For any $u, v$,*

$$|R(\sigma_{D_u}; D_u) - R(\sigma_{D_v}; D_v)| \leq |u - v|.$$

*Proof.* As $\sigma_{D_u}$ is computed by backward induction from Definition A.1, we have $R(\sigma_{D_u}; D_u) = \sigma_{D_u, 1} = \max\{u, \sigma_{D_u, 2}\}$. Moreover, since the distribution of boxes from $2, \ldots, n$ are the same, $\sigma_{D_u, i} = \sigma_{D_v, i}$ for all $i \geq 2$. For any $u \geq v$,

$$R(\sigma_{D_u}; D_u) - R(\sigma_{D_v}; D_v) = \max\{u, \sigma_{D_u, 2}\} - \max\{v, \sigma_{D_v, 2}\} \in [0, u - v],$$

where the inequality holds since max function is 1-Lipschitz and $\sigma_{D_u, 2} = \sigma_{D_v, 2}$. Thus, the lemma follows. $\qquad \square$

In the Prophet Inequality problem, the definition of $\widetilde{R}_i(\sigma; z)$ takes the same form as in Eq. (6), but uses the utility function $R$ defined in Eq. (9). Similarly, we follow Eq. (15) to define $\widetilde{R}_i(\sigma; x_{<i}, z)$, again using the utility function defined in Eq. (9).

**Lemma C.3.2.** *For all $t \in [T]$, $i \in [n]$, $\widetilde{R}_i(\sigma; z)$ is 1-Lipschitz and monotonically-increasing with respect to $z$ in Prophet Inequality problem.*

*Proof.* As the order of the boxes is fixed across time, we use Lemma C.3.1 to repeat a similar argument in Lemma 3.2.1 to complete the proof. The only difference is that conditioning on event that the learner opens box $i$, $\widetilde{R}_i(\sigma; x_{<i}, z)$ can be interpreted as the expected utility of running $\sigma_t$ on $(x_{i+1}, \ldots, x_n) \sim \mathcal{H}_{t, >i}$ with $x_i = z$ since $x_1, \ldots, x_{i-1}$ are discarded. $\qquad \square$

**Lemma C.3.3** (**Sharp Bound of Derivative for Prophet Inequality**)**.** *The Prophet Inequality problem satisfies*

$$\frac{\partial}{\partial z} \widetilde{R}_i(\sigma_t; z) = 0, \quad \forall z \in [0, \sigma_{t,i}).$$

*Proof.* For any $0 \leq v \leq u < \sigma_{t,i}$, we have that

$$\widetilde{R}_i(\sigma_t; u) - \widetilde{R}_i(\sigma_t; v) = \underset{x_{<i} \sim (D_1, \cdots, D_{i-1})}{\mathbb{E}} \left[ \widetilde{R}_i(\sigma_t; x_{<i}, u) - \widetilde{R}_i(\sigma_t; x_{<i}, v) \mid E_{\sigma_t, i} \right] = 0,$$

where the second equality uses the fact that if the learner opens box $i$ and the realized value is smaller than $\sigma_{t,i}$, then they will discard it and keep opening subsequent boxes. Thus, the lemma follows. $\qquad \square$

## C.4  Proof of $\widetilde{\mathcal{O}}(\sqrt{nT})$ Regret Bound for Pandora's Box

The analysis in this section conditions on the event that the concentration bounds of Lemma C.1.2 hold with respect to $D$ and $\mathcal{E}_t$ for all $t$. Recall from Section 3.2 that regret is bounded by

$$\mathrm{Reg}_T \leq \sum_{t=1}^{T}\sum_{i=1}^{n}\mathbb{E}\left[Q_{t,i}\cdot\underbrace{\left(\mathop{\mathbb{E}}_{z\sim\hat{\mathcal{E}}_{t,i}}\left[\widetilde{R}_i(\sigma_t;z)\right]-\mathop{\mathbb{E}}_{z\sim D_i}\left[\widetilde{R}_i(\sigma_t;z)\right]\right)}_{=:\,\mathtt{Term}_{t,i}}\right]. \tag{17}$$

Consider any given round $t$ and suppose, without loss of generality, that

$$\sigma_{t,1}\geq\sigma_{t,2}\geq\cdots\geq\sigma_{t,n}.$$

Since $\widetilde{R}_i(\sigma_t;z)$ is 1-Lipschitz with respect to $z$, the map $\widetilde{R}_i(\sigma_t;z)$ is absolutely continuous on $[0,1]$, which implies that it is differentiable almost everywhere in $[0,1]$. By the fundamental theorem of calculus, for any $x\in[0,1]$, we have that $\widetilde{R}_i(\sigma_t;x)=\widetilde{R}_i(\sigma_t;0)+\int_0^x\frac{\partial}{\partial z}\widetilde{R}_i(\sigma_t;z)\,dz$. Therefore, we have that

$$\begin{aligned}
\mathtt{Term}_{t,i} &= \mathop{\mathbb{E}}_{z\sim\hat{\mathcal{E}}_{t,i}}\left[\widetilde{R}_i(\sigma_t;z)\right]-\mathop{\mathbb{E}}_{z\sim D_i}\left[\widetilde{R}_i(\sigma_t;z)\right]\\
&= \mathop{\mathbb{E}}_{z\sim\hat{\mathcal{E}}_{t,i}}\left[\widetilde{R}_i(\sigma_t;0)+\int_0^z\frac{\partial}{\partial t}\widetilde{R}_i(\sigma_t;t)\,dt\right]-\mathop{\mathbb{E}}_{z\sim D_i}\left[\widetilde{R}_i(\sigma_t;0)+\int_0^z\frac{\partial}{\partial t}\widetilde{R}_i(\sigma_t;t)\,dt\right]\\
&= \mathop{\mathbb{E}}_{z\sim\hat{\mathcal{E}}_{t,i}}\left[\int_0^1\frac{\partial}{\partial t}\widetilde{R}_i(\sigma_t;t)\mathbb{I}\{z\geq t\}\,dt\right]-\mathop{\mathbb{E}}_{z\sim D_i}\left[\int_0^1\frac{\partial}{\partial t}\widetilde{R}_i(\sigma_t;t)\mathbb{I}\{z\geq t\}\,dt\right]\\
&= \int_0^1\left(1-F_{\hat{\mathcal{E}}_{t,i}}(z)\right)\frac{\partial}{\partial z}\widetilde{R}_i(\sigma_t;z)\,dz-\int_0^1\left(1-F_{D_i}(z)\right)\frac{\partial}{\partial z}\widetilde{R}_i(\sigma_t;z)\,dz.
\end{aligned}$$

According to the analysis in Section 3.2, we bound

$$\mathtt{Term}_{t,i}\leq\underbrace{\int_{\sigma_{t,i+1}}^{1}\left|F_{D_i}(z)-F_{\hat{\mathcal{E}}_{t,i}}(z)\right|\,dz}_{A_{t,i}}+\underbrace{\int_0^{\sigma_{t,i+1}}\left(F_{D_i}(z)-F_{\hat{\mathcal{E}}_{t,i}}(z)\right)\frac{\partial}{\partial z}\widetilde{R}_i(\sigma_t;z)\,dz}_{B_{t,i}}.$$

Now, we complete the bounds on $A_{t,i}$ and $B_{t,i}$.

**Bounding $A_{t,i}$.** We have

$$\begin{aligned}
A_{t,i} &\leq \int_{\sigma_{t,i+1}}^{1}\left|F_{D_i}(z)-F_{\mathcal{E}_{t,i}}(z)\right|\,dz+\int_{\sigma_{t,i+1}}^{1}\left|F_{\mathcal{E}_{t,i}}(z)-F_{\hat{\mathcal{E}}_{t,i}}(z)\right|\,dz\\
&\leq \mathcal{O}\left(\frac{L}{m_{t,i}}+\int_{\sigma_{t,i+1}}^{1}\sqrt{\frac{F_{D_i}(z)(1-F_{D_i}(z))L}{m_{t,i}}}\,dz+\int_{\sigma_{t,i+1}}^{1}\sqrt{\frac{F_{\mathcal{E}_{t,i}}(z)(1-F_{\mathcal{E}_{t,i}}(z))L}{m_{t,i}}}\,dz\right)\\
&\leq \mathcal{O}\left(\frac{L}{m_{t,i}}+\sqrt{\frac{(1-F_{D_i}(\sigma_{t,i+1}))L}{m_{t,i}}}+\sqrt{\frac{(1-F_{\mathcal{E}_{t,i}}(\sigma_{t,i+1}))L}{m_{t,i}}}\right)\\
&\leq \mathcal{O}\left(\sqrt{\frac{(1-F_{D_i}(\sigma_{t,i+1}))L}{m_{t,i}}}+\frac{L}{m_{t,i}}\right), \tag{18}
\end{aligned}$$

where the second inequality bounds $|F_{D_i}(z)-F_{\mathcal{E}_{t,i}}(z)|$ by Lemma C.1.2 and bounds $|F_{\mathcal{E}_{t,i}}(z)-F_{\hat{\mathcal{E}}_{t,i}}(z)|$ by the construction Eq. (1), and the third inequality bounds $1-F_{D_i}(z)\leq 1-F_{D_i}(\sigma_{t,i+1})$ (similar for $1-F_{\mathcal{E}_{t,i}}(z)$) as $z\geq\sigma_{t,i+1}$. The last inequality follows from the following reasoning:

$$\begin{aligned}
&\sqrt{\frac{(1-F_{\mathcal{E}_{t,i}}(\sigma_{t,i+1}))L}{m_{t,i}}}\\
&\leq \sqrt{\frac{(1-F_{D_i}(\sigma_{t,i+1}))L+L\left|F_{\mathcal{E}_{t,i}}(\sigma_{t,i+1})-F_{D_i}(\sigma_{t,i+1})\right|}{m_{t,i}}}
\end{aligned}$$

$$\leq \sqrt{\frac{(1 - F_{D_i}(\sigma_{t,i+1}))L}{m_{t,i}}} + \sqrt{\frac{L\left|F_{\mathcal{E}_{t,i}}(\sigma_{t,i+1}) - F_{D_i}(\sigma_{t,i+1})\right|}{m_{t,i}}}$$

$$\leq \sqrt{\frac{(1 - F_{D_i}(\sigma_{t,i+1}))L}{m_{t,i}}} + \frac{\left|F_{\mathcal{E}_{t,i}}(\sigma_{t,i+1}) - F_{D_i}(\sigma_{t,i+1})\right|}{2} + \frac{L}{m_{t,i}}$$

$$\leq \mathcal{O}\left(\sqrt{\frac{(1 - F_{D_i}(\sigma_{t,i+1}))L}{m_{t,i}}} + \frac{L}{m_{t,i}}\right), \tag{19}$$

where the third inequality uses $\sqrt{2ab} \leq a + b$ for any $a, b \geq 0$, and the last inequality bounds $\left|F_{\mathcal{E}_{t,i}}(\sigma_{t,i+1}) - F_{D_i}(\sigma_{t,i+1})\right|$ by Lemma C.1.2.

**Bounding $B_{t,i}$.** By Lemma 3.2.2, we bound

$$B_{t,i} \leq \int_0^{\sigma_{t,i+1}} \left|F_{D_i}(z) - F_{\hat{\mathcal{E}}_{t,i}}(z)\right| \frac{\sqrt{\prod_{j<i} F_{D_j}(z)}\sqrt{\prod_{j<i} F_{D_j}(z)}}{Q_{t,i}} \, dz$$

$$\leq \int_0^{\sigma_{t,i+1}} \left(\left|F_{D_i}(z) - F_{\mathcal{E}_{t,i}}(z)\right| + \left|F_{\mathcal{E}_{t,i}}(z) - F_{\hat{\mathcal{E}}_{t,i}}(z)\right|\right) \sqrt{\frac{\prod_{j<i} F_{D_j}(z)}{Q_{t,i}}} \, dz, \tag{20}$$

where the second inequality bounds $\prod_{j<i} F_{D_j}(z) \leq \prod_{j<i} F_{D_j}(\sigma_{t,i+1}) \leq \prod_{j<i} F_{D_j}(\sigma_{t,i}) = Q_{t,i}$ for all $z < \sigma_{t,i+1}$.

On the one hand, we use Lemma C.1.2 to bound for any $z$

$$\left|F_{D_i}(z) - F_{\mathcal{E}_{t,i}}(z)\right| \leq \mathcal{O}\left(\sqrt{\frac{F_{D_i}(z)(1 - F_{D_i}(z))L}{m_{t,i}}} + \frac{L}{m_{t,i}}\right) \leq \mathcal{O}\left(\sqrt{\frac{(1 - F_{D_i}(z))L}{m_{t,i}}} + \frac{L}{m_{t,i}}\right).$$

On the other hand, we use the construction of optimistic distribution $\hat{\mathcal{E}}_{t,i}$ in Eq. (1) to show for any $z$

$$\left|F_{\mathcal{E}_{t,i}}(z) - F_{\hat{\mathcal{E}}_{t,i}}(z)\right| \leq \mathcal{O}\left(\sqrt{\frac{(1 - F_{\mathcal{E}_{t,i}}(z))L}{m_{t,i}}} + \frac{L}{m_{t,i}}\right) \leq \mathcal{O}\left(\sqrt{\frac{(1 - F_{D_i}(z))L}{m_{t,i}}} + \frac{L}{m_{t,i}}\right),$$

where the last inequality repeats the same argument in Eq. (19).

Plugging the two bounds above into Eq. (20), we have that

$$B_{t,i} \leq \mathcal{O}\left(\int_0^{\sigma_{t,i+1}} \left(\sqrt{\frac{(1 - F_{D_i}(z))L}{m_{t,i}}} + \frac{L}{m_{t,i}}\right) \sqrt{\frac{\prod_{j<i} F_{D_j}(z)}{Q_{t,i}}} \, dz\right)$$

$$= \mathcal{O}\left(\int_0^{\sigma_{t,i+1}} \left(\sqrt{\frac{\prod_{j<i} F_{D_j}(z)(1 - F_{D_i}(z))L}{Q_{t,i}m_{t,i}}} + \sqrt{\frac{\prod_{j<i} F_{D_j}(z)}{Q_{t,i}}} \frac{L}{m_{t,i}}\right) \, dz\right)$$

$$\leq \mathcal{O}\left(\frac{L}{m_{t,i}} + \int_0^{\sigma_{t,i+1}} \left(\sqrt{\frac{\prod_{j<i} F_{D_j}(z)(1 - F_{D_i}(z))L}{Q_{t,i}m_{t,i}}}\right) \, dz\right)$$

$$\leq \mathcal{O}\left(\frac{L}{m_{t,i}} + \int_0^1 \left(\sqrt{\frac{\prod_{j<i} F_{D_j}(z)(1 - F_{D_i}(z))L}{Q_{t,i}m_{t,i}}}\right) \, dz\right),$$

where the second inequality follows from

$$\sqrt{\frac{\prod_{j<i} F_{D_j}(z)}{Q_{t,i}}} \leq \sqrt{\frac{\prod_{j<i} F_{D_j}(\sigma_{t,i+1})}{Q_{t,i}}} \leq \sqrt{\frac{\prod_{j<i} F_{D_j}(\sigma_{t,i})}{Q_{t,i}}} = 1$$

based on the fact that $z \leq \sigma_{t,i+1}$ and $Q_{t,i} = \prod_{j<i} F_{D_j}(\sigma_{t,i})$.

**Regret for round $t$.** From the bounds of $A_{t,i}$ and $B_{t,i}$, the regret at round $t$ is bounded as

$$\text{Reg}(t) \leq \sum_{i \in [n]} \text{Term}_{t,i} \leq \mathcal{O}\left( \mathbb{E}\left[ \sum_{i \in [n]} Q_{t,i}\sqrt{\frac{(1 - F_{D_i}(\sigma_{t,i+1}))L}{m_{t,i}}} \right.\right.$$

$$\left.\left. + \sum_{i \in [n]} \int_0^1 \sqrt{\frac{\prod_{j<i} F_{D_j}(z)(1 - F_{D_i}(z))LQ_{t,i}}{m_{t,i}}}\, dz + \frac{LQ_{t,i}}{m_{t,i}} \right] \right).$$

The first term is bounded as follows:

$$\mathbb{E}\left[ \sum_{i \in [n]} Q_{t,i}\sqrt{\frac{(1 - F_{D_i}(\sigma_{t,i+1}))L}{m_{t,i}}} \right]$$

$$\leq \mathbb{E}\left[ \sqrt{\sum_{i \in [n]} Q_{t,i}(1 - F_{D_i}(\sigma_{t,i+1}))}\sqrt{\sum_{i \in [n]} \frac{LQ_{t,i}}{m_{t,i}}} \right] \leq \mathbb{E}\left[ \sqrt{\sum_{i \in [n]} \frac{LQ_{t,i}}{m_{t,i}}} \right] . =,$$

where the first inequality uses the Cauchy–Schwarz inequality, and the second inequality follows from the the fact that

$$\sqrt{\sum_{i \in [n]} Q_{t,i}(1 - F_{D_i}(\sigma_{t,i+1}))} = \sqrt{\sum_{i \in [n]} \prod_{j<i} F_{D_j}(\sigma_{t,i})(1 - F_{D_i}(\sigma_{t,i+1}))}$$

$$\leq \sqrt{\sum_{i \in [n]} \prod_{j<i} F_{D_j}(\sigma_{t,j+1})(1 - F_{D_i}(\sigma_{t,i+1}))}$$

$$= \sqrt{\sum_{i \in [n]} \left( \prod_{j<i} F_{D_j}(\sigma_{t,j+1}) - \prod_{j\leq i} F_{D_j}(\sigma_{t,j+1}) \right)}$$

$$\leq 1.$$

Here, the first inequality holds since the descending sorted assumption on $\sigma_t$ gives $\sigma_{t,j+1} \geq \sigma_{t,i}$ for all $j < i$, and the last inequality follows from the fact that the summation forms a telescoping sum.

To bound the second term, we show that

$$\sum_{i \in [n]} \int_0^1 \sqrt{\frac{\prod_{j<i} F_{D_j}(z)(1 - F_{D_i}(z))LQ_{t,i}}{m_{t,i}}}\, dz$$

$$= \int_0^1 \sum_{i \in [n]} \sqrt{\prod_{j<i} F_{D_j}(z)(1 - F_{D_i}(z))}\sqrt{\frac{LQ_{t,i}}{m_{t,i}}}\, dz$$

$$\leq \int_0^1 \sqrt{\sum_{i \in [n]} \prod_{j<i} F_{D_j}(z)(1 - F_{D_i}(z))}\sqrt{\sum_{i \in [n]} \frac{LQ_{t,i}}{m_{t,i}}}\, dz$$

$$\leq \sqrt{\sum_{i \in [n]} \frac{LQ_{t,i}}{m_{t,i}}},$$

where the first inequality uses the Cauchy–Schwarz inequality, and the last inequality follows from the fact that $\sum_{i \in [n]} \prod_{j<i} F_{D_j}(z)(1 - F_{D_i}(z))$ forms a telescoping sum.

Combining all bounds above, we have

$$\text{Reg}(t) \leq \mathcal{O}\left( \mathbb{E}\left[ \sqrt{\sum_{i \in [n]} \frac{LQ_{t,i}}{m_{t,i}}} + \frac{LQ_{t,i}}{m_{t,i}} \right] \right).$$

**Algorithm 4** Proposed algorithm for Prophet Inequality
___
**Input**: confidence $\delta \in (0, 1)$, horizon $T$.
**Initialize**: open all boxes once and observe rewards. Set $m_{1,i} = 1$ for all $i \in [n]$.
**for** $t = 1, 2, \ldots, T$ **do**

> For each $i \in [n]$, construct an optimistic distribution $\hat{\mathcal{E}}_{t,i}$ according to Eq. (1).
> Use Definition A.1 with distribution $\hat{\mathcal{E}}_t$ to compute $\sigma_t = (\sigma_{t,1}, \ldots, \sigma_{t,n})$.
> Run Algorithm 3 with $\sigma_t$ to open a set of boxes, denoted by $\mathcal{B}_t$ and observe rewards $\{v_{t,i}\}_{i \in \mathcal{B}_t}$.
> Update counters $m_{t+1,i} = m_{t,i} + 1, \forall i \in \mathcal{B}_t$ and $m_{t+1,i} = m_{t,i}, \forall i \notin \mathcal{B}_t$.

___

**Summing** $\text{Reg}(t)$ **over all** $t$. The Cauchy–Schwarz inequality gives

$$
\text{Reg}_T \leq \mathcal{O}\left( \mathbb{E}\left[ \sum_{t=1}^{T} \left( \sqrt{\sum_{i \in [n]} \frac{LQ_{t,i}}{m_{t,i}}} + \frac{LQ_{t,i}}{m_{t,i}} \right) \right] \right)
$$

$$
\leq \mathcal{O}\left( \mathbb{E}\left[ \sqrt{T \sum_{i \in [n]} \sum_{t=1}^{T} \frac{LQ_{t,i}}{m_{t,i}}} + \sum_{t=1}^{T} \frac{LQ_{t,i}}{m_{t,i}} \right] \right)
$$

$$
\leq \mathcal{O}\left( \sqrt{T \sum_{i \in [n]} \mathbb{E}\left[ \sum_{t=1}^{T} \frac{LQ_{t,i}}{m_{t,i}} \right]} + \mathbb{E}\left[ \sum_{t=1}^{T} \frac{LQ_{t,i}}{m_{t,i}} \right] \right), \tag{21}
$$

where the last inequality uses Jensen's inequality.

Finally, it remains to bound $\mathbb{E}\left[ \sum_{t=1}^{T} LQ_{t,i}/m_{t,i} \right]$. Let $I_{t,i} = \mathbb{I}\{E_{\sigma_t,i}\}$.

$$
\mathbb{E}\left[ \sum_{t=1}^{T} \sum_{i \in [n]} \frac{LQ_{t,i}}{m_{t,i}} \right] = \mathbb{E}\left[ \sum_{t=1}^{T} \sum_{i \in [n]} \frac{L\mathbb{E}_t[I_{t,i}]}{m_{t,i}} \right] = \mathbb{E}\left[ \sum_{t=1}^{T} \sum_{i \in [n]} \frac{LI_{t,i}}{m_{t,i}} \right] \leq \mathcal{O}\left( nL \log(T) \right), \tag{22}
$$

where the second equality holds due to $m_{t,i}$ is deterministic given history before round $t$.

Combining the bound $\mathbb{E}\left[ \sum_{t=1}^{T} LQ_{t,i}/m_{t,i} \right] = \mathcal{O}(nL \log T)$ with Eq. (21) completes the proof of Theorem 3.1.3.

## C.5 $\widetilde{\mathcal{O}}(\sqrt{nT})$ Regret Bound for Prophet Inequality

The proposed algorithm for the Prophet Inequality problem is shown in Algorithm 4.

**Theorem C.5.1.** *For the Prophet Inequality problem, choosing $\delta = T^{-1}$ for Algorithm 4 ensures*

$$
\text{Reg}_T = \mathcal{O}\left( \sqrt{nT \log(T) \log(nT)} + n \log(T) \log(nT) \right).
$$

*Proof.* The analysis conditions on the event that the concentration bounds of Lemma C.1.2 hold with respect to $D$ and $\mathcal{E}_t$ for all $t$. As shown by Guo et al. [2021], the Prophet Inequality satisfies the same monotonicity as Pandora's Box. Thus, the analysis follows the same approach as used in the Pandora's Box problem to bound $\text{Reg}_T \leq \sum_{t=1}^{T} \sum_{i=1}^{n} \mathbb{E}[Q_{t,i} \cdot \text{Term}_{t,i}]$ where $\text{Term}_{t,i}$ has the same definition as in Eq. (4) with a new utility function $R$ defined in Eq. (9).

As shown in Lemma C.3.2, the Prophet Inequality problem has both monotonicity and smoothness properties. Therefore, repeating the same analysis in Section 3.2 (splitting the integral on $\sigma_{t,i}$ instead of $\sigma_{t,i+1}$) gives

$$
\text{Term}_{t,i} \leq \int_{\sigma_{t,i}}^{1} \left| F_{D_i}(z) - F_{\hat{\mathcal{E}}_{t,i}}(z) \right| dz + \int_{0}^{\sigma_{t,i}} \left( F_{D_i}(z) - F_{\hat{\mathcal{E}}_{t,i}}(z) \right) \frac{\partial}{\partial z} \widetilde{R}_i(\sigma_t; z) \, dz
$$

$$
= \underbrace{\int_{\sigma_{t,i}}^{1} \left| F_{D_i}(z) - F_{\hat{\mathcal{E}}_{t,i}}(z) \right| dz}_{A_{t,i}},
$$

where the equality applies Lemma C.3.3. We use the same argument as in Eq. (18) to bound $A_{t,i}$: indeed,

$$A_{t,i} \leq \mathcal{O}\left(\sqrt{\frac{(1 - F_{D_i}(\sigma_{t,i}))L}{m_{t,i}}} + \frac{L}{m_{t,i}}\right).$$

Thus, the regret at round $t$ is bounded by

$$\text{Reg}(t) \leq \mathcal{O}\left(\sum_{i \in [n]} Q_{t,i} \sqrt{\frac{(1 - F_{D_i}(\sigma_{t,i}))L}{m_{t,i}}} + \sum_{i \in [n]} \frac{Q_{t,i}L}{m_{t,i}}\right).$$

In particular, we bound

$$\sum_{i \in [n]} Q_{t,i} \sqrt{\frac{(1 - F_{D_i}(\sigma_{t,i}))L}{m_{t,i}}}$$

$$\leq \sqrt{\sum_{i \in [n]} Q_{t,i}(1 - F_{D_i}(\sigma_{t,i}))} \sqrt{\sum_{i \in [n]} \frac{Q_{t,i}L}{m_{t,i}}}$$

$$= \sqrt{\sum_{i \in [n]} \prod_{j < i} F_{D_j}(\sigma_{t,j})(1 - F_{D_i}(\sigma_{t,i}))} \sqrt{\sum_{i \in [n]} \frac{Q_{t,i}L}{m_{t,i}}} \leq \sqrt{\sum_{i \in [n]} \frac{Q_{t,i}L}{m_{t,i}}},$$

where the equality holds since $Q_{t,i} = \prod_{j < i} F_{D_j}(\sigma_{t,j})$ in the Prophet Inequality setting.

By repeating the same argument as in Eq. (21) and Eq. (22), the proof of Theorem C.5.1 is complete. $\square$

## C.6 High-Probability Regret Bounds

The regret metric now is

$$\text{Reg}'_T := \sum_{t=1}^{T}(R(\sigma_D; D) - R(\sigma_t; D)). \tag{23}$$

Following the same argument in Section 3.2, we arrive at Eq. (4) without expectation. As $\sigma_t$ and $H_{t,i}$ are deterministic given history, we have

$$R(\sigma_t; H_{t,i-1}) - R(\sigma_t; H_{t,i}) = \mathbb{E}_t[R(\sigma_t; H_{t,i-1}) - R(\sigma_t; H_{t,i})],$$

where $\mathbb{E}_t[\cdot]$ is the conditional expectation given history before round $t$. Then, we can follow the same analysis to arrive at $\sqrt{\sum_{i,t} \frac{Q_{t,i}}{m_{t,i}}}$. To handle this, let $M_{t,i} = \frac{Q_{t,i} - I_{t,i}}{m_{t,i}}$, and $\{M_{t,i}\}_t$ is martingale difference sequence. By Freedman's inequality, with probability at least $1 - \delta$, we have $\sum_t M_{t,i} \leq \sqrt{2V \log(1/\delta)} + \log(1/\delta)$ where $V = \sum_{t=1}^{T} \mathbb{E}_t[M_{t,i}^2]$. Notice that $V = \frac{Q_{t,i} - Q_{t,i}^2}{m_{t,i}} \leq \frac{Q_{t,i}}{m_{t,i}}$. Thus, we have $\sum_t M_{t,i} \leq \sqrt{2 \log(1/\delta) \sum_t Q_{t,i}/m_{t,i}} + \log(1/\delta) \leq \frac{1}{2} \sum_t Q_{t,i}/m_{t,i} + 2 \log(1/\delta)$ where the last step uses the AM-GM inequality. Rearranging it gives $\sum_{t=1}^{T} \frac{Q_{t,i}}{m_{t,i}} \leq O(\log(1/\delta) + \sum_{t=1}^{T} \frac{I_{t,i}}{m_{t,i}})$. Therefore, by choosing $\delta$ properly and applying a union bound, with high probability, the regret is bounded by $\widetilde{O}(\sqrt{nT})$.

## C.7 Extension to Subgaussian Rewards

For each box, if the reward distribution is sub-Gaussian, then our algorithms still work with minor modifications. Specifically, if the reward distribution of box $i$ is $K_i$-sub-Gaussian, one can set a range $[-K_i\sqrt{2\log(2T^2 n)}, K_i\sqrt{2\log(2T^2 n)}]$. Then, the algorithm updates parameters only if the received reward of each box $i$ is within range of $[-K_i\sqrt{2\log(2T^2 n)}, K_i\sqrt{2\log(2T^2 n)}]$. On those rounds, the algorithm and its regret analysis coincide exactly with the bounded-reward setting. Meanwhile, by a union bound on sub-Gaussian tails, the probability that any reward ever exceeds its range is at most $1/T$. Therefore, its contribution to the expected regret is only $\widetilde{O}(1)$.

# D Omitted Details of Contextual Linear Setting

## D.1 Preliminaries

**Lemma D.1.1** (DKW Inequality). *Given a natural number $N$, let $X_1, \dots, X_N$ be i.i.d. samples from distribution $D$ with cumulative distribution function $F_D(\cdot)$. Let $F_{\mathcal{E}}(\cdot)$ be the associated empirical distribution function $\mathcal{E}$ such that for all $x \in \mathbb{R}$, we define $F_{\mathcal{E}}(x) := \frac{1}{N} \sum_{i=1}^{N} \mathbf{1}\{X_i \leq x\}$. For every $\varepsilon > 0$,*

$$
\mathbb{P}\left( \sup_{x \in \mathbb{R}} \left| F_{\mathcal{E}}(x) - F_D(x) \right| > \varepsilon \right) \leq 2\exp\left(-2N\varepsilon^2\right).
$$

**Lemma D.1.2.** *Let $D = (D_1, \dots, D_n)$ be a product distribution, and for each $i$, let $\mathcal{E}_{N_i, i}$ be the empirical distribution built from $N_i$ i.i.d. samples of $D_i$. With probability at least $1 - \delta/2$, for any $i \in [n]$, any $N_i \in [T]$,*

$$
\sup_{z \in \mathbb{R}} \left| F_{D_i}(z) - F_{\mathcal{E}_{N_i, i}}(z) \right| \leq \sqrt{\frac{\log(4Tn\delta^{-1})}{2N_i}}. \tag{24}
$$

*Proof.* We first fix any $i$ and $N_i$ and apply Lemma D.1.1. Then, the lemma follows by using a union bound over all $i \in [n]$ and $N_i \in [T]$. $\qquad\square$

**Lemma D.1.3.** *With probability at least $1 - \delta/2$, for any $t > 0$ and any $i \in [n]$,*

$$
\left\| \theta_i - \widehat{\theta}_{t,i} \right\|_{V_{t,i}} \leq \alpha_\delta, \quad \text{where} \quad \alpha_\delta = 1 + \sqrt{2\log(2n/\delta) + d\log\left(1 + \frac{T}{d}\right)}.
$$

*Proof.* Fix any $i \in [n]$. Since the noise distribution of each box is $\frac{1}{4}$-subgaussian, and $\{x_{t,i}\}_t$ is a $\mathbb{R}^d$-valued stochastic process such that $x_{t,i}$ is measurable with respect to the history, [Lattimore and Szepesvári, 2020, Theorem 20.5] directly gives the claimed result for the fixed $i$. Using a union bound over all $i \in [n]$ completes the proof. $\qquad\square$

**Lemma D.1.4** (**Elliptical Potential Lemma** (see, e.g., Lemma 11 in [Abbasi-Yadkori et al., 2011])). *Let $I \in \mathbb{R}^{d \times d}$ be an identity matrix and a sequence of vectors $a_1, \dots, a_N \in \mathbb{R}^d$ with $\|a_i\|_2 \leq 1$ for all $i \in [N]$. Let $V_t = I + \sum_{s \leq t} a_s a_s^\top$. Then, for all $N \in \mathbb{N}$, we have*

$$
\sum_{t=1}^{N} \min\left\{ 1, \|a_t\|_{V_{t-1}^{-1}}^2 \right\} \leq 2d\log\left(1 + \frac{N}{d}\right).
$$

**Definition D.1.5** (**Definition of $\widetilde{D}_t$**). *Let $\widetilde{D}_t = (\widetilde{D}_{t,1}, \dots, \widetilde{D}_{t,n})$ where each $\widetilde{D}_{t,i}$ is an empirical distribution which assigns $m_{t,i}$ i.i.d. noise samples from $D_i$ with equal probability mass $1/m_{t,i}$.*

**Definition D.1.6** (**"Nice" event**). *Let $\mathscr{E}$ be the nice event that the concentration bounds in Lemma D.1.2 for $D, \widetilde{D}_t$ hold for all $t \in [T]$, and the concentration bounds in Lemma D.1.3 hold simultaneously.*

**Lemma D.1.7.** *Suppose that $\mathscr{E}$ holds. For any context $x \in \mathcal{X}$ and any round $t \in [T]$,*

$$
\left| \left\langle x, \widehat{\theta}_{t,i} - \theta_i \right\rangle \right| \leq \beta_i^{t+1}(x).
$$

*Proof.* For any $t, i, x$, one can show that

$$
\left| \left\langle x, \widehat{\theta}_{t,i} - \theta_i \right\rangle \right| \leq \|x\|_{V_{t,i}^{-1}} \left\| \theta_i - \widehat{\theta}_{t,i} \right\|_{V_{t,i}} \leq \alpha_\delta \|x\|_{V_{t,i}^{-1}} = \beta_i^{t+1}(x),
$$

where the first inequality follows from the Cauchy–Schwarz inequality and the second inequality uses Lemma D.1.3. $\qquad\square$

## D.2  Algorithms and $\widetilde{\mathcal{O}}(nd\sqrt{T})$ Bounds for Pandora's Box and Prophet Inequality

The omitted algorithm for both the Pandora's Box and Prophet Inequality problems is presented in Algorithm 5. Unlike the non-contextual case (see Eq. (1)), we use a slightly different construction for the optimistic distribution. This is because the Bernstein-type construction requires access to the CDF of an empirical distribution based on i.i.d. samples, which is unavailable in our setting. As a result, we use the following approach instead.

**Optimistic distribution** $\hat{\mathcal{E}}_{t,i}$**.** Let $L = \frac{1}{2}\log(4nT/\delta)$. The CDF of $\hat{\mathcal{E}}_{t,i}$ is constructed as follows:

$$F_{\hat{\mathcal{E}}_{t,i}}(x) = \begin{cases} 1, & x = 1; \\ \max\left\{0, F_{\mathcal{E}_{t,i}}(x) - \sqrt{\frac{L}{m_{t,i}}}\right\}, & 0 \le x < 1. \end{cases} \tag{25}$$

Now, we prove Lemma 4.1 which shows that the optimistic distribution $\hat{\mathcal{E}}_t$ stochastically dominates $D_t$ for all $t$.

**Lemma D.2.1** (Restatement of Lemma 4.1). *Suppose that $\mathscr{E}$ holds. For all $t \in [T]$, $\hat{\mathcal{E}}_t \succeq_{SD} D_t$.*

*Proof.* For this proof, we show $\hat{\mathcal{E}}_{t,i} \succeq_{\text{SD}} D_{t,i}$ for all $i, t$. Fix a box $i \in [n]$ and a round $t \in [T]$. If $c \ge 1$, then

$$\mathbb{P}_{X \sim \hat{\mathcal{E}}_{t,i}}(X \le c) = \mathbb{P}_{X \sim \hat{D}_{t,i}}(X \le c) = 1.$$

For any $c < 1$, we have that

$$\mathbb{P}_{X \sim \hat{\mathcal{E}}_{t,i}}(X \le c) = \max\left\{0, \mathbb{P}_{X \sim \mathcal{E}_{t,i}}(X \le c) - \sqrt{\frac{L}{m_{t,i}}}\right\}$$

$$\le \max\left\{0, \mathbb{P}_{Z \sim D_{t,i}}(Z \le c)\right\}$$

$$= \mathbb{P}_{Z \sim D_{t,i}}(Z \le c),$$

The inequality holds due to the following reason:

$$\mathbb{P}_{X \sim \mathcal{E}_{t,i}}(X \le c) - \sqrt{\frac{L}{m_{t,i}}}$$

$$= \frac{1}{m}\sum_{j=1}^{m}\mathbb{I}\left\{\eta_{t_i(j),i} + \mu_{t_i(j),i} - \text{LCB}_i^t(\widehat{\theta}_{t-1,i}, x_{t_i(j),i}) + \text{UCB}_i^t(\widehat{\theta}_{t-1,i}, x_{t,i}) \le c\right\} - \sqrt{\frac{L}{m_{t,i}}}$$

$$\le \frac{1}{m}\sum_{j=1}^{m}\mathbb{I}\left\{\eta_{t_i(j),i} + \text{UCB}_i^t(\widehat{\theta}_{t-1,i}, x_{t,i}) \le c\right\} - \sqrt{\frac{L}{m_{t,i}}}$$

$$\le \frac{1}{m_{t,i}}\sum_{j=1}^{m}\mathbb{I}\left\{\eta_{t_i(j),i} \le c - \mu_{t,i}\right\} - \sqrt{\frac{L}{m_{t,i}}}$$

$$\le \mathbb{P}_{X \sim D_i}(X \le c - \mu_{t,i})$$

$$= \mathbb{P}_{Z \sim D_{t,i}}(Z \le c),$$

where the first inequality follows from the facts that $\mu_{t_i(j),i} - \text{LCB}_i^t(\widehat{\theta}_{t-1,i}, x_{t_i(j),i}) \ge 0$, the second inequality uses the fact that $\text{UCB}_i^t(\widehat{\theta}_{t-1,i}, x_{t,i}) \ge \mu_{t,i}$, and the last inequality follows from the concentration bound given by $\mathscr{E}$.

Conditioning on the "nice" event, the argument holds for all $i$ and $t$. The proof is complete. $\square$

Next, we present the following lemma, whose proof is in the next subsection.

**Lemma D.2.2.** *Suppose that $\mathscr{E}$ holds. For both the Pandora's Box and Prophet Inequality problems, we have that*

$$\text{Reg}_T \le \sum_{t=1}^{T}\sum_{i=1}^{n}\mathcal{O}\left(\mathbb{E}\left[Q_{t,i}\left(\sqrt{\frac{L}{m_{t,i}}} + \frac{1}{m_{t,i}}\sum_{j=1}^{m_{t,i}}\left|z_{t_i(j),i}^t - \hat{z}_{t_i(j),i}^t\right|\right)\right]\right).$$

---

**Algorithm 5** Proposed algorithm for contextual linear model

---

**Input**: confidence $\delta \in (0,1)$, horizon $T$.

**Initialize**: observe context $\{x_{0,i}\}_{i\in[n]}$ for each box, open each box, observe rewards $\{v_{0,i}\}_{i\in[n]}$, and update counters $\{m_{1,i}\}_{i\in[n]}$ and parameters $\{\widehat{\theta}_{0,i}\}_{i\in[n]}$ where $\widehat{\theta}_{0,i} = \left(I + x_{0,i}x_{0,i}^\top\right)^{-1} x_{0,i}v_{0,i}$.

**for** $t = 1,2,\ldots,T$ **do**

> Observe contexts $(x_{t,1},\ldots,x_{t,n})$ and compute $\widehat{\mu}_{t,i} = \left\langle \widehat{\theta}_{t-1,i}, x_{t,i} \right\rangle$ for each $i \in [n]$.
>
> For each $i \in [n]$, use Eq. (25) to construct an empirical distribution $\widehat{\mathcal{E}}_{t,i}$ by using samples
>
> $$\left\{ v_{s,i} - \text{LCB}_i^t(\widehat{\theta}_{t-1,i}, x_{s,i}) + \text{UCB}_i^t(\widehat{\theta}_{t-1,i}, x_{t,i}) \right\}_{s<t:i\in\mathcal{B}_s}.$$
>
> **if** *Pandora's Box* **then**
>
> > Compute $\sigma_t = (\sigma_{t,1},\ldots,\sigma_{t,n})$ where $\sigma_{t,i}$ is the solution of $\mathbb{E}_{x\sim\widehat{\mathcal{E}}_{t,i}}[(x - \sigma_{t,i})_+] = c_i$.
> > Run Algorithm 1 with $\sigma_t$ to open a set of boxes, denoted by $\mathcal{B}_t$ to observe rewards $\{v_{t,i}\}_{i\in\mathcal{B}_t}$.
>
> **else if** *Prophet Inequality* **then**
>
> > Use Definition A.1 with distribution $\widehat{\mathcal{E}}_t$ to compute $\sigma_t = (\sigma_{t,1},\ldots,\sigma_{t,n})$.
> > Run Algorithm 3 with $\sigma_t$ to open a set of boxes, denoted by $\mathcal{B}_t$ to observe rewards $\{v_{t,i}\}_{i\in\mathcal{B}_t}$.
>
> Update counters $m_{t+1,i} = m_{t,i} + 1, \forall i \in \mathcal{B}_t$ and $m_{t+1,i} = m_{t,i}, \forall i \notin \mathcal{B}_t$.
> Update estimate for each $i \in \mathcal{B}_t$:
>
> $$V_{t,i} = I + \sum_{s\leq t:i\in\mathcal{B}_s} x_{s,i}x_{s,i}^\top \quad \text{where} \quad \widehat{\theta}_{t,i} = V_{t,i}^{-1} \sum_{s\leq t:i\in\mathcal{B}_s} x_{s,i}v_{s,i}.$$

---

With Lemma D.2.2 in hand, we are now ready to prove Theorem 4.2.

*Proof of Theorem 4.2.* The analysis conditions on event $\mathscr{E}$ which is defined in Definition D.1.6 and occurs with probability at least $1 - \delta$. Recall that $I_{t,i} = \mathbb{I}\{E_{\sigma_t,i}\}$. For the first term, we take the iterated expectation and use the facts that $Q_{t,i} = \mathbb{E}_t[I_{t,i}]$ and $m_{t,i}$ is deterministic given the history to get

$$\sum_{t=1}^T \sum_{i=1}^n \mathbb{E}\left[Q_{t,i}\sqrt{\frac{L}{m_{t,i}}}\right] = \sum_{t=1}^T \sum_{i=1}^n \mathbb{E}\left[I_{t,i}\sqrt{\frac{L}{m_{t,i}}}\right] \leq \mathcal{O}(n\sqrt{TL}).$$

Next, we bound the summation of the second term. To this end, we show that

$$\left| z_{t_i(j),i}^t - \hat{z}_{t_i(j),i}^t \right|$$
$$= \min\left\{1, \left| z_{t_i(j),i}^t - \hat{z}_{t_i(j),i}^t \right|\right\}$$
$$= \min\left\{1, \left| \eta_{t_i(j),i} + \mu_{t,i} - \min\left\{1, v_{t_i(j),i} - \text{LCB}_i^t(\widehat{\theta}_{t-1,i}, x_{t_i(j),i}) + \text{UCB}_i^t(\widehat{\theta}_{t-1,i}, x_{t,i})\right\}\right|\right\}$$
$$\leq \min\left\{1, \left| v_{t_i(j),i} - \text{LCB}_i^t(\widehat{\theta}_{t-1,i}, x_{t_i(j),i}) + \text{UCB}_i^t(\widehat{\theta}_{t-1,i}, x_{t,i}) - \eta_{t_i(j),i} - \mu_{t,i}\right|\right\}$$
$$\leq \min\left\{1, \left| v_{t_i(j),i} - \text{LCB}_i^t(\widehat{\theta}_{t-1,i}, x_{t_i(j),i}) - \eta_{t_i(j),i}\right| + \left| \text{UCB}_i^t(\widehat{\theta}_{t-1,i}, x_{t,i}) - \mu_{t,i}\right|\right\}$$
$$= \min\left\{1, \left| \mu_{t_i(j),i} - \text{LCB}_i^t(\widehat{\theta}_{t-1,i}, x_{t_i(j),i})\right| + \left| \text{UCB}_i^t(\widehat{\theta}_{t-1,i}, x_{t,i}) - \mu_{t,i}\right|\right\}$$
$$= \min\left\{1, \left| \left\langle x_{t_i(j),i}, \theta_i - \widehat{\theta}_{t-1,i}\right\rangle + \beta_i^t(x_{t_i(j),i})\right| + \left| \left\langle x_{t,i}, \theta_i - \widehat{\theta}_{t-1,i}\right\rangle + \beta_i^t(x_{t,i})\right|\right\}$$
$$\leq \min\left\{1, 2\beta_i^t(x_{t_i(j),i}) + 2\beta_i^t(x_{t,i})\right\}$$
$$\leq \min\left\{1, 2\beta_i^t(x_{t_i(j),i})\right\} + \min\left\{1, 2\beta_i^t(x_{t,i})\right\}, \tag{26}$$

where the first equality holds due to $z_{t_i(j),i}^t, \hat{z}_{t_i(j),i}^t \in [0,1]$, the first inequality uses $|b - \min\{1,a\}| \leq |a - b|$ for $b \in [0,1]$, and the third inequality follows from Lemma D.1.7.

Using Eq. (26), we have that

$$\sum_{t=1}^{T}\sum_{i=1}^{n}\mathbb{E}\left[\frac{Q_{t,i}}{m_{t,i}}\sum_{j=1}^{m_{t,i}}\left|z_{t_i(j),i}^{t}-\hat{z}_{t_i(j),i}^{t}\right|\right]$$

$$\leq 2\sum_{t=1}^{T}\sum_{i=1}^{n}\mathbb{E}\left[Q_{t,i}\left(\min\left\{1,\beta_i^t(x_{t,i},\delta)\right\}+\frac{1}{m_{t,i}}\sum_{j=1}^{m_{t,i}}\min\left\{1,\beta_i^t(x_{t_i(j),i},\delta)\right\}\right)\right].$$

For the second term on the right hand side, we have that

$$\sum_{t=1}^{T}\sum_{i=1}^{n}\mathbb{E}\left[\frac{Q_{t,i}}{m_{t,i}}\sum_{j=1}^{m_{t,i}}\beta_i^t(x_{t_i(j),i},\delta)\right]$$

$$\leq\alpha_\delta\sum_{t=1}^{T}\sum_{i=1}^{n}\mathbb{E}\left[\frac{Q_{t,i}}{m_{t,i}}\sum_{j=1}^{m_{t,i}}\min\left\{1,\left\|x_{t_i(j),i}\right\|_{V_{t-1,i}^{-1}}\right\}\right]$$

$$\leq\alpha_\delta\sum_{t=1}^{T}\sum_{i=1}^{n}\mathbb{E}\left[\frac{Q_{t,i}}{m_{t,i}}\sum_{j=1}^{m_{t,i}}\min\left\{1,\left\|x_{t_i(j),i}\right\|_{V_{t_i(j)-1,i}^{-1}}\right\}\right]$$

$$\leq\alpha_\delta\sum_{t=1}^{T}\sum_{i=1}^{n}\mathbb{E}\left[\frac{Q_{t,i}}{m_{t,i}}\sqrt{m_{t,i}\sum_{j=1}^{m_{t,i}}\min\left\{1,\left\|x_{t_i(j),i}\right\|_{V_{t_i(j)-1,i}^{-1}}^{2}\right\}}\right]$$

$$\leq\mathcal{O}\left(\alpha_\delta\sum_{t=1}^{T}\sum_{i=1}^{n}\mathbb{E}\left[Q_{t,i}\sqrt{\frac{d\log(1+T/d)}{m_{t,i}}}\right]\right)$$

$$\leq\mathcal{O}\left(\alpha_\delta\cdot n\sqrt{dT\log(1+T/d)}\right),$$

where the second inequality follows from $V_{t-1,i}\succeq V_{t_i(j)-1,i}$ for all $t$ and $i$, the third inequality uses Cauchy–Schwarz inequality, and the fourth inequality follows from Lemma D.1.4.

Analogously, we take the iterated expectation and use the fact that $Q_{t,i}=\mathbb{E}_t[I_{t,i}]$ and $\|x_{t,i}\|_{V_{t-1,i}^{-1}}$ is deterministic given the history to conclude that

$$\sum_{t=1}^{T}\sum_{i=1}^{n}\mathbb{E}\left[Q_{t,i}\beta_i^t(x_{t,i},\delta)\right]$$

$$\leq\alpha_\delta\sum_{t=1}^{T}\sum_{i=1}^{n}\mathbb{E}\left[Q_{t,i}\min\left\{1,\|x_{t,i}\|_{V_{t-1,i}^{-1}}\right\}\right]$$

$$=\alpha_\delta\sum_{i=1}^{n}\mathbb{E}\left[\sum_{t=1}^{T}I_{t,i}\min\left\{1,\|x_{t,i}\|_{V_{t-1,i}^{-1}}\right\}\right]$$

$$=\alpha_\delta\sum_{i=1}^{n}\mathbb{E}\left[\sum_{j=1}^{m_{T,i}}\min\left\{1,\|x_{t_i(j),i}\|_{V_{t_i(j)-1,i}^{-1}}\right\}\right]$$

$$\leq\alpha_\delta\sum_{i=1}^{n}\mathbb{E}\left[\sqrt{m_{T,i}\sum_{j=1}^{m_{T,i}}\min\left\{1,\|x_{t_i(j),i}\|_{V_{t_i(j)-1,i}^{-1}}^{2}\right\}}\right]$$

$$\leq\mathcal{O}\left(\alpha_\delta\cdot n\sqrt{dT\log(1+T/d)}\right).$$

Combining the bounds above completes the proof of Theorem 4.2. $\qquad\square$

## D.3 Proof of Lemma D.2.2

We follow a similar analysis of the non-contextual case to focus on the regret bound at each round and then sum them up together. Let us consider a round $t$. For Pandora's Box problem, we assume

W.L.O.G. that $\sigma_{t,1} \geq \sigma_{t,2} \geq \cdots \geq \sigma_{t,n}$. Similar to the non-contextual analysis, we define for all $t, i$

$$\mathcal{H}_{t,0} = \hat{\mathcal{E}}_t, \quad \text{and} \quad \mathcal{H}_{t,i} = \left( D_{t,1}, \cdots, D_{t,i}, \hat{\mathcal{E}}_{t,i+1}, \cdots, \hat{\mathcal{E}}_{t,n} \right).$$

We have $\mathcal{H}_{t,n} = D_t$ based on this definition. As Lemma 4.1 ensures the stochastic dominance and both problems enjoy a certain monotonicity, we follow the same analysis in Section 3.2 to show that

$$\text{Reg}(t) = \mathbb{E}\left[ R(\sigma_D; D) - R(\sigma_t; D) \right]$$

$$\leq \sum_{i=1}^{n} \mathbb{E}\left[ Q_{t,i} \underbrace{\mathbb{E}_{x \sim \mathcal{H}_{t,i-1}} \left[ R\left( \sigma_t; x \right) \mid E_{\sigma_t, i} \right] - \mathbb{E}_{y \sim \mathcal{H}_{t,i}} \left[ R\left( \sigma_t; y \right) \mid E_{\sigma_t, i} \right]}_{\text{Term}_{t,i}} \right].$$

Now, we define

- $\hat{\mathcal{H}}_{t,i} := \left( D_{t,1}, \cdots, D_{t,i-1}, \hat{D}_{t,i}, \hat{\mathcal{E}}_{t,i+1}, \cdots, \hat{\mathcal{E}}_{t,n} \right)$, and
- $\widetilde{\mathcal{H}}_{t,i} := \left( D_{t,1}, \cdots, D_{t,i-1}, \mathcal{E}_{t,i}, \hat{\mathcal{E}}_{t,i+1}, \cdots, \hat{\mathcal{E}}_{t,n} \right)$.

Based on the definitions of $\mathcal{H}_{t,i}, \hat{\mathcal{H}}_{t,i}, \widetilde{\mathcal{H}}_{t,i}$, we compute the following decomposition:

$$\text{Term}_{t,i} = \mathbb{E}_{x \sim \mathcal{H}_{t,i-1}} \left[ R\left( \sigma_t; x \right) \mid E_{\sigma_t, i} \right] - \mathbb{E}_{y \sim \mathcal{H}_{t,i}} \left[ R\left( \sigma_t; y \right) \mid E_{\sigma_t, i} \right]$$

$$= \underbrace{\mathbb{E}_{x \sim \mathcal{H}_{t,i-1}} \left[ R\left( \sigma_t; x \right) \mid E_{\sigma_t, i} \right] - \mathbb{E}_{x \sim \widetilde{\mathcal{H}}_{t,i}} \left[ R\left( \sigma_t; x \right) \mid E_{\sigma_t, i} \right]}_{(I)}$$

$$+ \underbrace{\mathbb{E}_{x \sim \widetilde{\mathcal{H}}_{t,i}} \left[ R\left( \sigma_t; x \right) \mid E_{\sigma_t, i} \right] - \mathbb{E}_{x \sim \hat{\mathcal{H}}_{t,i}} \left[ R\left( \sigma_t; x \right) \mid E_{\sigma_t, i} \right]}_{(II)}$$

$$+ \underbrace{\mathbb{E}_{x \sim \hat{\mathcal{H}}_{t,i}} \left[ R\left( \sigma_t; x \right) \mid E_{\sigma_t, i} \right] - \mathbb{E}_{x \sim \mathcal{H}_{t,i}} \left[ R\left( \sigma_t; x \right) \mid E_{\sigma_t, i} \right]}_{(III)}.$$

In the remaining analysis, we retain the same definition of $\widetilde{R}_i(\sigma_t; z)$ as in Eq. (6). The analysis conditions on this history before round $t$, and thus $D_t$ is a product distribution.

**Bounding (I).** By using the fact that $E_{\sigma_t, i}$ is independent of $i$-th box's reward, we can bound

$$(I) = \mathbb{E}_{x \sim \mathcal{H}_{t,i-1}} \left[ R\left( \sigma_t; x \right) \mid E_{\sigma_t, i} \right] - \mathbb{E}_{x \sim \widetilde{\mathcal{H}}_{t,i}} \left[ R\left( \sigma_t; x \right) \mid E_{\sigma_t, i} \right]$$

$$= \mathbb{E}_{z \sim \hat{\mathcal{E}}_{t,i}} \left[ \widetilde{R}_i(\sigma_t; z) \right] - \mathbb{E}_{z \sim \mathcal{E}_{t,i}} \left[ \widetilde{R}_i(\sigma_t; z) \right]$$

$$= \int_0^1 \left( 1 - F_{\hat{\mathcal{E}}_{t,i}}(z) \right) \frac{\partial}{\partial z} \widetilde{R}_i(\sigma_t; z) \, dz - \int_0^1 \left( 1 - F_{\mathcal{E}_{t,i}}(z) \right) \frac{\partial}{\partial z} \widetilde{R}_i(\sigma_t; z) \, dz$$

$$= \int_0^1 \left( F_{\mathcal{E}_{t,i}}(z) - F_{\hat{\mathcal{E}}_{t,i}}(z) \right) \frac{\partial}{\partial z} \widetilde{R}_i(\sigma_t; z) \, dz$$

$$\leq \int_0^1 \left| F_{\mathcal{E}_{t,i}}(z) - F_{\hat{\mathcal{E}}_{t,i}}(z) \right| \, dz$$

$$\leq \mathcal{O}\left( \sqrt{\frac{L}{m_{t,i}}} \right),$$

where the first inequality follows from the fact that $\widetilde{R}_i(\sigma_t; z)$ is 1-Lipschitz for both Pandora's Box and Prophet Inequality problems (see Lemma 3.2.1 and Lemma C.3.2, respectively), and the last inequality uses the construction of $\hat{\mathcal{E}}_{t,i}$ in Eq. (25).

**Bounding (II).** One can show that

$$
\begin{aligned}
\text{(II)} &= \mathop{\mathbb{E}}_{x \sim \widetilde{\mathcal{H}}_{t,i}} \left[ R\left(\sigma_t; x\right) \mid E_{\sigma_t,i} \right] - \mathop{\mathbb{E}}_{x \sim \widehat{\mathcal{H}}_{t,i}} \left[ R\left(\sigma_t; x\right) \mid E_{\sigma_t,i} \right] \\
&= \mathop{\mathbb{E}}_{z \sim \mathcal{E}_{t,i}} \left[ \widetilde{R}_i(\sigma_t; z) \right] - \mathop{\mathbb{E}}_{z \sim \hat{D}_{t,i}} \left[ \widetilde{R}_i(\sigma_t; z) \right] \\
&= \frac{1}{m_{t,i}} \sum_{j=1}^{m_{t,i}} \widetilde{R}_i \left( \sigma_t; \hat{z}_{t_i(j),i}^t \right) - \frac{1}{m_{t,i}} \sum_{j=1}^{m_{t,i}} \widetilde{R}_i \left( \sigma_t; z_{t_i(j),i}^t \right) \\
&\leq \frac{1}{m_{t,i}} \sum_{j=1}^{m_{t,i}} \left| z_{t_i(j),i}^t - \hat{z}_{t_i(j),i}^t \right|,
\end{aligned}
$$

where the inequality again the fact that $\widetilde{R}_i(\sigma_t; z)$ is 1-Lipschitz for both Pandora's Box and Prophet Inequality problems.

**Bounding (III).** Then, we have

$$
\begin{aligned}
\text{(III)} &= \mathop{\mathbb{E}}_{x \sim \widehat{\mathcal{H}}_{t,i}} \left[ R\left(\sigma_t; x\right) \mid E_{\sigma_t,i} \right] - \mathop{\mathbb{E}}_{x \sim \mathcal{H}_{t,i}} \left[ R\left(\sigma_t; x\right) \mid E_{\sigma_t,i} \right] \\
&= \mathop{\mathbb{E}}_{z \sim \hat{D}_{t,i}} \left[ \widetilde{R}_i(\sigma_t; z) \right] - \mathop{\mathbb{E}}_{z \sim D_{t,i}} \left[ \widetilde{R}_i(\sigma_t; z) \right] \\
&= \int_0^1 \left( F_{D_{t,i}}(z) - F_{\hat{D}_{t,i}}(z) \right) \frac{\partial}{\partial z} \widetilde{R}_i(\sigma_t; z) \, dz \\
&\leq \int_0^1 \left| F_{D_{t,i}}(z) - F_{\hat{D}_{t,i}}(z) \right| dz \\
&= \int_0^1 \left| F_{D_i}(z - \mu_{t,i}) - F_{\widetilde{D}_{t,i}}(z - \mu_{t,i}) \right| dz \\
&\leq \mathcal{O}\left( \sqrt{\frac{L}{m_{t,i}}} \right),
\end{aligned}
$$

where the first inequality follows from the fact that $\frac{\partial}{\partial z} \widetilde{R}_i(\sigma_t; z) \in [-1, 1]$ due to 1-Lipschitzness, and the last inequality follows from the concentration bound given by $\mathscr{E}$.

**Putting together.** Combining all bounds above, we have

$$
\text{Reg}(t) \leq \sum_{i=1}^n \mathcal{O}\left( \mathbb{E}\left[ Q_{t,i} \left( \sqrt{\frac{L}{m_{t,i}}} + \frac{1}{m_{t,i}} \sum_{j=1}^{m_{t,i}} \left| z_{t_i(j),i}^t - \hat{z}_{t_i(j),i}^t \right| \right) \right] \right).
$$

Summing over all $t \in [T]$, we complete the proof of Lemma D.2.2.

