# OpenReview forum: "Improved Regret and Contextual Linear Extension for Pandora's Box and Prophet Inequality"
_NeurIPS.cc/2025/Conference — NeurIPS 2025 poster_

### Official Review · Reviewer_3xhg · 2025-06-23

**Clarity:** 3
**Significance:** 3
**Originality:** 2
**Rating:** 4
**Confidence:** 4

**Summary:**

This work improves the regret guarantee for the online learning problem in the Pandora's box problem from $\tilde{O}(n\sqrt{T})$ to $\tilde{O}(\sqrt{nT})$, via a Bernstein-type construction for the optimistic construction.
The paper further extends the problem to the contextual case with linear rewards, and achieves an $\tilde{O}(nd\sqrt{T})$ regret in that problem, where $d$ is the dimension of the context.

**Questions:**

- Could the authors explain in detail how their use of Bernstein inequality is different from Guo at al. (2019)?
- Could the authors address why the similar algorithm gives different reliance on $n$ for the regret bound on the non-contextual and contextual problems?

**Ethical Concerns:**

["NO or VERY MINOR ethics concerns only"]

**Final Justification:**

I will keep my score.

**Limitations:**

Yes.

**Paper Formatting Concerns:**

No.

**Quality:**

3

**Strengths And Weaknesses:**

The results of this paper are solid, and the paper is written well, striking a good balance between intuitions and technical details.
However, I think some discussions are still required to further improve the readability of this paper.

First, Bernstein-type solutions are given in Guo et al. (2019) to solve the sample complexity for auctions and other related problems.
This paper mentions that the use in analysis is substantially different from that of Guo et al. (2019), but not in detail.
As for me, the idea of using Bernstein inequality to construct stochastically dominated empirical distributions is similar.

Second, it is interesting and also natural to ask why the dependence on the number of boxes $n$ is different for the non-contextual and contextual problems in this paper, but the authors do not discuss this point.

---

> ### Author Rebuttal · Authors · 2025-07-29
>
> We thank the reviewer for the valuable feedback. Below we address your concerns.
>
> ---
> **W1 and Q1:** This paper mentions that the use in analysis is substantially different from that of Guo et al. (2019), but not in detail. As for me, the idea of using Bernstein inequality to construct stochastically dominated empirical distributions is similar. Could the authors explain in detail how their use of Bernstein inequality is different from Guo et al. (2019)?
>
> **A:** Our use differs significantly in both the algorithmic perspective and the analysis.
>
>
> - **Algorithmic perspective.** Since Guo et al. (2019) focuses on the sample complexity, they first collect sufficient samples over all the distributions of the boxes to construct empirical distributions, and then output a near-optimal policy from *pessimistic distributions*, which reallocate the probability mass of empirical distributions toward *lower outcomes*.
> This is *exactly the opposite of our optimistic scheme*, where we shift mass to *higher outcomes*.
> As we focus on the regret minimization, the algorithm needs to balance exploration and exploitation.
> To this end, our algorithm uses optimistic distributions with Bernstein-type adjustment, which encourages the algorithm to adaptively explore the distributions of the boxes.
>
>
>
> - **Analytical perspective.** The analysis of (Guo et al. 2019) requires *sufficient and equal* amount of samples for each box. However, since our algorithm adaptively explores boxes, the  sample size varies across boxes, and some boxes may only have a small sample size.
> We therefore conduct a finer‑grained study of the exploration process and show that a Bernstein‑type concentration bound naturally captures its adaptiveness and delivers the desired regret guarantees.
>
>
>
> ---
> **W2 and Q2:** it is interesting and also natural to ask why the dependence on the number of boxes $n$ is different for the non-contextual and contextual problems in this paper, but the authors do not discuss this point. Could the authors address why the similar algorithm gives different reliance on $n$ for the regret bound on the non-contextual and contextual problems?
>
> **A:** The different dependence on $n$ arises primarily because in the contextual setting, our analysis needs to bound the estimation error of context-dependent mean, which leads to a linear dependence on $n$.
> Unlike the non-contextual case where the reward samples are i.i.d., in the contextual case the mean reward varies with the context, and the observed samples are not identically distributed across rounds.
> To ensure stochastic dominance, we construct an optimistic distribution by shifting the sample values in an optimistic manner. The regret introduced by this value-shifting scales linearly with $n$, and the technique that yields the $\widetilde{O}(\sqrt{nT})$ bound in the non-contextual setting, does not  extend to the contextual case.

---

> > ### Comment · Reviewer_3xhg · 2025-08-05
> >
> > Thank the authors for their reply. I appreciate that, and highly recommend them to add the above discussions into their final version.

---

### Official Review · Reviewer_oZNk · 2025-06-30

**Clarity:** 3
**Significance:** 3
**Originality:** 3
**Rating:** 5
**Confidence:** 4

**Summary:**

The authors study the Pandora's Box problem in a semi-bandit feedback setting, where the learning agent observes the realized rewards of the open boxes in a round. The first main contribution is regret bound of $\tilde{O}(\sqrt{nT})$, which improves upon the state-of-the-art. The improvement is brought about by an application of a Bernstein-type concentration inequality for a tighter upper confidence bound in the OFU-based algorithm, as well as a more refined sensitivity analysis on the error term $Term_{i, t}$ in Section 3.2. The authors also provide an improvement in a linear contextual generalization, and they improve the regret bound scaling from $T^{5/6}$ ($T$ is the number of rounds) to $T^{1/2}$.

**Questions:**

In addition to the question raised in Weakness point 2, I have an additional question and an additional suggestion:

Question: It appears to me that the construction of the empirical distributions in the linear contextual case is different from the non-contextual case. Can I check that if I apply the algorithm in Section 4 on the non-contextual case (1 feature dimension per box), do we end up with $O(n^2 \sqrt{T})$ regret, or is there some better way?

Suggestion: The authors' model could have some connection to the cascading bandit model:

https://arxiv.org/abs/1502.02763

While the two models involve semi-bandit feedback depending on the ordering of basic arms/boxes, they are quite different in terms of the reward function. Nevertheless, I believe the above work is worth mentioning.

**Ethical Concerns:**

["NO or VERY MINOR ethics concerns only"]

**Final Justification:**

I have read the response, and is satisfied with the response on the lower bound. I recommend the authors to provide more details that the lower bounds apply also to the authors' feedback setting to make things clearer.

**Limitations:**

Yes

**Quality:**

4

**Strengths And Weaknesses:**

Strengths:

1. I find the overall theoretical contributions substantial and worthwhile. Although the algorithm design is following the well-established optimism-in-face-of-uncertainty principle, the overall careful analysis that leads to $\sqrt{n}$ is still an interesting technical contributions. The sensitivity analysis is tailored to the Pandora's Box setting, and appears novel in my opinion.

2. The sketch proof in Section 3.2 is quite clearly written, and will be interesting to researchers to explore further generalizations.

3. In the linear contextual case, the improvement in regret bound is substantial.

Weaknesses:

1. The boundedness assumption on the random noise in the linear contextual case seems to be a limiting assumption. In the traditional linear bandits, it suffices to assume a bounded mean reward, while the noise can have an unbounded supported as long as it is sub-Guassian. What is the fundamental barrier that forbids such a generalization.

2. The authors assert that their regret bound of $O(\sqrt{n T})$ is tight, due to the regret lower bound of $\Omega(\sqrt{nT})$ in Gatmiry et al. [16]. My understanding is that Gatmiry et al. [16] focuses on the purely bandit feedback case, meaning that after each round, the learning agent only observes the net utility gained, but does not observe the individual rewards of the opened boxes. Can the authors confirm that the lower bound in Gatmiry et al. [16] also applies in the authors' semi-bandit feedback setting? If yes, the authors should provide more explanations. If it is the otherwise, the authors should emphasize on the difference between the feedback model.

---

> ### Author Rebuttal · Authors · 2025-07-29
>
> We thank the reviewer for the valuable and positive feedback. Below we address your concerns.
>
> ---
> **W1:** The boundedness assumption on the random noise in the linear contextual case seems to be a limiting assumption. In the traditional linear bandits, it suffices to assume a bounded mean reward, while the noise can have an unbounded supported as long as it is sub-Guassian.
>
> **A:** In fact, this assumption can be relaxed.
> For each box, if the noise distribution is sub-Gaussian, then, the reward distribution is also sub-Gaussian, and our algorithms still work with minor modifications. Specifically, if the reward distribution of box $i$ is $K_i$-sub-Gaussian, one can set a range $[-K_i \sqrt{2\log(2T^2n)},K_i\sqrt{2\log(2T^2n)}]$.
> Then, the algorithm updates parameters only if the received reward of each box $i$ is within range of $[-K_i \sqrt{2\log(2T^2n)},K_i\sqrt{2\log(2T^2n)}]$.
> On those rounds, the algorithm and its regret analysis coincide exactly with the bounded‑reward setting.
> Meanwhile, by a union bound on sub-Gaussian tails, the probability that any reward ever exceeds its range is at most
> $1/T$.   Therefore, its contribution to the expected regret is only $\widetilde{O}(1)$.
>
> ---
> **W2:** Does $\Omega(\sqrt{nT})$ lower bound in Gatmiry et al. [16] apply here?
>
> **A:** The $\Omega(\sqrt{nT})$ lower bound in Gatmiry et al. [16] indeed applies to our setting since their lower bound holds even in the full-information setting, thereby applying to ours. Specifically, Gatmiry et al., prove $\Omega(\sqrt{nT})$ regret lower bound by using $\Omega(n/\epsilon^2)$ lower bound for sample complexity. To see this, we here reiterate their proof idea. For the full-information feedback, if an online algorithm achieves $o(\sqrt{nT})$ regret, then one can use online-to-batch conversion to get a $\epsilon$-optimal policy with $o(n/\epsilon^2)$ sample complexity, which contradicts to $\Omega(n/\epsilon^2)$ lower bound.
>
>
> ---
> **Q1:** Can I check that if I apply the algorithm in Section 4 on the non-contextual case (1 feature dimension per box), do we end up with $O(n^2\sqrt{T})$ regret bound?
>
>
> **A:** Our contextual setting uses a separate parameter $\theta_i$ for each box $i$, which subsumes the non-contextual case as a special instance. In the non-contextual case, it suffices to set the feature dimension $d=1$, leading to a $\widetilde{O}(n\sqrt{T})$ regret bound. While this is worse than the $\widetilde{O}(\sqrt{nT})$ bound of our dedicated non-contextual algorithm, the linear dependence on $n$ arises from the estimation error of context-dependent means. The technique used to obtain the tighter $\widetilde{O}(\sqrt{nT})$ bound does not carry over to the contextual setting. Whether the $\widetilde{O}(nd\sqrt{T})$ bound is tight remains an open question for future work.
>
>
> ---
> **Q2:** The authors' model could have some connection to the cascading bandit model: ``Kveton et al., Cascading Bandits: Learning to Rank in the Cascade Model, ICML 2015''. While the two models involve semi-bandit feedback depending on the ordering of basic arms/boxes, they are quite different in terms of the reward function. Nevertheless, I believe the above work is worth mentioning.
>
> **A:** We thank the reviewer for highlighting this interesting paper. We will make sure to include this reference in the revision.

---

### Official Review · Reviewer_vRPG · 2025-07-02

**Clarity:** 4
**Significance:** 3
**Originality:** 3
**Rating:** 5
**Confidence:** 4

**Summary:**

This paper studies an online variant of the Pandora’s box problem with $n$ boxes. In this problem, the player sequentially plays T instances of the Pandora’s box problems, one in each round. The assumption is that the reward distributions are unknown to the player but remain fixed throughout the horizon. In this setting, the authors prove an $O(\sqrt{nT})$ regret bound against the best policy in hindsight, thereby improving the existing $O(n\sqrt{T})$ regret bound. Their algorithm first constructs an optimistic version of the empirical distributions for each box, and then it computes the thresholds by using Weitzman’s method applied to the optimistic distributions. They obtain improved regret by using a sharper bound for a regret upper bound obtained earlier by [Agarwal et al].

**Questions:**

1. Is it possible to establish $O(\sqrt{nT})$ regret bound with high probability? The paper only proves this bound in expectation.
2. The authors assume that the reward distributions have a bounded support. Can the results be extended to distributions with unbounded support?
3. For completeness, it is better to briefly reproduce the arguments from [2] leading to Eqn (4).

**Ethical Concerns:**

["NO or VERY MINOR ethics concerns only"]

**Final Justification:**

The authors have addressed my concerns. I'll keep my positive score.

**Quality:**

3

**Strengths And Weaknesses:**

The refined analysis is technically novel and interesting. The authors also extend their result to a linear contextual setting and the prophet inequality. The paper is well-written, easy to follow, and the related work has been discussed adequately.

---

> ### Author Rebuttal · Authors · 2025-07-29
>
> We thank the reviewer for positive feedback and your acknowledgment on our contributions. Below we address your concerns.
>
> ---
> **Q1:** Is it possible to establish $\widetilde{O}(\sqrt{nT})$ regret bound with high probability?
>
>
> **A:** Yes. We can obtain a high-probability bound of $\widetilde{O}(\sqrt{nT})$ under regret metric $\sum_{t=1}^T (R(\sigma\_{D};D)-R(\sigma\_{t};D))$ via two modifications in our analysis. First, LHS of Eq(4) now has no expectation, but since $\sigma_t$ and $H_{t,i}$ are deterministic given history, we have $R(\sigma\_t;H\_{t,i-1})-R(\sigma\_t;H\_{t,i}) = \mathbb{E}\_t [R(\sigma_t;H_{t,i-1})-R(\sigma_t;H_{t,i})]$ where $\mathbb{E}\_t[\cdot]$ is the conditional expectation given history before round $t$. Then, we can follow the same analysis to arrive at $\sqrt{\sum_{i,t} \frac{Q_{t,i}}{m_{t,i}} }$.
> To handle this, let $M_{t,i}=\frac{Q_{t,i}-I_{t,i}}{m_{t,i}}$, and $\\{M_{t,i}\\}\_{t}$ is martingale difference sequence. By Freedman's inequality, with probability at least $1-\delta$, we have $\sum_{t} M_{t,i} \leq \sqrt{2V \log(1/\delta)} + \log(1/\delta)$ where $V= \sum_{t=1}^T \mathbb{E}\_t [M_{t,i}^2 ]$. Notice that $V =\frac{ Q_{t,i} -Q_{t,i}^2   }{m_{t,i}} \leq \frac{ Q_{t,i}  }{m_{t,i}}$. Thus, we have $\sum_{t} M_{t,i} \leq \sqrt{ 2\log(1/\delta)\sum_{t}Q_{t,i}/m_{t,i}  }+\log(1/\delta) \leq  \frac{1}{2}\sum_{t}Q_{t,i}/m_{t,i}+ 2\log(1/\delta) $ where the last step uses the AM-GM inequality. Rearranging it gives
> $\sum_{t=1}^T \frac{Q_{t,i}}{m_{t,i}} \leq O (\log(1/\delta)+\sum_{t=1}^T \frac{I_{t,i}}{m_{t,i}} )$.
> Therefore, by choosing $\delta$ properly and applying a union bound, with high probability, the regret is bounded by $\widetilde{O}(\sqrt{nT})$.
>
> ---
> **Q2:** The authors assume that the reward distributions have a bounded support. Can the results be extended to distributions with unbounded support?
>
>
> **A:** For an arbitrary unbounded distribution, we believe the online Pandora's Box is intractable without further assumption.
> The main hardness here is that a box may yield an extremely large reward with a very small probability. For example, consider two instances in which there is only one box cost $1$. In instance $I_1$, the box generates reward $2^T$ with probability $2^{-T+1}$, and generates reward $0$ otherwise. In the other instance $I_2$, the box generates the same reward with probability $2^{-T-1}$ and $0$ otherwise. Then, the learner needs to decide whether to open the box. The optimal strategy in $I_1$ always opens the box, whereas the optimal strategy in $I_2$ never opens it. However, distinguishing these two instances are statistically hard within $T$ rounds, and thus the online Pandora's Box with arbitrary unbounded distribution is intractable.
>
>
> However, if the reward distribution of each box is subgaussian, then, our algorithms still work with minor modifications. Specifically, if the reward distribution of box $i$ is $K_i$-subgaussian, one can set a range $[-K_i \sqrt{2\log(2T^2n)},K_i\sqrt{2\log(2T^2n)}]$.
> Then, the algorithm updates parameters only if the received reward of each box $i$ is within range of $[-K_i \sqrt{2\log(2T^2n)},K_i\sqrt{2\log(2T^2n)}]$.
> On those rounds, the algorithm and its regret analysis coincide exactly with the bounded‑reward setting.
> Meanwhile, by a union bound on subgaussian tails, the probability that any reward ever exceeds its range is at most
> $1/T$. Therefore, its contribution to the expected regret is only $\widetilde{O}(1)$.
>
>
> ---
> **Q3:** For completeness, it is better to briefly reproduce the arguments from [2] leading to Eqn (4).
>
> **A:** Thank you for the suggestion. We will reproduce those arguments in the next version.

---

> > ### Comment · Reviewer_vRPG · 2025-08-01
> > **Thanks for your response**
> >
> > I would like to thank the authors for their responses. The responses did not significantly change my perspective on the paper, thus, I will keep my current evaluation.

---

### Official Review · Reviewer_HDSt · 2025-07-02

**Clarity:** 1
**Significance:** 3
**Originality:** 3
**Rating:** 4
**Confidence:** 2

**Summary:**

This paper studies the online Pandora’s Box problem and proposes an algorithm that achieves a regret bound matching the known lower bound and improves over Agarwal et al. in the non-contextual setting. The authors then extend their approach to both the contextual linear setting and online Prophet Inequality problem. The key idea is to mimic Weitzman’s optimal policy by constructing optimistic empirical distributions that stochastically dominate the unknown true distributions. This allows the algorithm to compute reservation values using Weitzman’s thresholding rule, while encouraging exploration.

**Questions:**

Why the contextual-linear setting is defined on a  [−1/4,1/4] support?

Why the principle of optimism under uncertainty is helpful for improving the regret guarantee in this work?

What is the weak benchmark considered by Gergatsouli and Tzamos? Similarly, what is the optimal policy benchmark considered in Gatmiry et al.? What is the benchmark used in this work for both the non-contextual and contextual cases?

**Ethical Concerns:**

["NO or VERY MINOR ethics concerns only"]

**Final Justification:**

After reading the responses, I agree the technical contribution is significant but can be stated more clearly. The writing also needs much improvement.

**Limitations:**

The limitations in the assumptions of the contextual linear setting compared to real-world motivations could be mentioned.

**Paper Formatting Concerns:**

No major formatting issues.

**Quality:**

2

**Strengths And Weaknesses:**

Strengths:

This paper tackles the online Pandora’s Box problem and proposes a novel algorithm that improves over prior work in both regret performance and generality. By leveraging optimistic empirical distributions to mimic Weitzman’s optimal threshold-based policy, the authors match the known minimax regret lower bound in the non-contextual setting, improving over the prior regret bound from Agarwal et al. The theoretical contributions are technically sound and show clear improvement over existing methods, and the analysis is rigorous with detailed proofs included.

Weaknesses:

The main weaknesses of this paper lie in clarity and structure, which make the contributions harder to interpret than necessary:

Some terms (e.g., full-information feedback, magnitude of the utility function) lack explanations.

Several naming choices could be improved: e.g., “Term” in Equation (4) is vague and could be replaced with something more meaningful.

The thresholds in Weitzman’s algorithm can be referred to more meaningful standard terminology such as reservation values, Weitzman’s indices or Gittins indices that are widely used in prior work.

The paper does not clearly distinguish between previously known and newly introduced formulations and results. For example, Section 3 should be titled "non-contextual online Pandora’s Box" to reflect the setting.

Theorem statements and algorithm descriptions lack sufficient context. For example, Algorithm 2 is used for the online problem, but is simply labeled for “Pandora’s Box”, which could be misread as referring to the classic setting. Similarly, Algorithm 1 is essentially Weitzman’s original policy for the classic setting, but this is never explicitly stated — and could be deferred to the appendix to prioritize space for the new contributions like Algorithm 5. Parameters such as delta are not referred with meaningful names in all theorem statements.

The problem formulation is fragmented: the assumptions, utility function definitions, and cumulative regret notions are introduced in separate places, making it difficult for readers to see the overall setup. A unified and clearly structured formulation section, distinguishing old and new contributions, would greatly improve clarity. It is also unclear why the contextual-linear setting is defined on a  [−1/4,1/4] support — this choice is not motivated or explained.

Moreover, the benchmark definitions used for regret are not clearly stated across settings. For example, the paper mentions that Gergatsouli and Tzamos consider a weak benchmark—but what precisely is that benchmark? Similarly, what is the optimal policy benchmark considered in Gatmiry et al.? What is the benchmark used in this work for both the non-contextual and contextual cases?

Finally, the proofs in the main text could be better structured by emphasizing high-level intuition and key steps, rather than presenting long sequences of inequalities that would be more appropriate in the appendix.

---

> ### Author Rebuttal · Authors · 2025-07-29
>
> We thank the reviewer for the valuable feedback. Below we address your concerns.
>
> ---
> **W1:** Improvement of clarity and structure.
>
> **A:** Thank you for the suggestions; we will edit the paper accordingly.  In particular, we will better emphasize the  intuition for the proofs in the main text and defer technical details to the appendix.
> That said, several of
> your comments are very minor and can be easily addressed. In fact, it is worth noting that all other reviewers commented and agreed that the paper is well-written. While we appreciate your suggestions and will revise accordingly, we sincerely hope that you could consider raising the score given that the main weakness you pointed out pertains to clarity and structure.
>
> ---
> **W2:** Algorithm 1 is essentially Weitzman’s original policy for the classic setting, but this is never explicitly stated.
>
> **A:** Algorithm 1 is a general version of Weitzman's algorithm since it accepts any threshold vector as an input. Algorithm 1 coincides with the classical Weitzman algorithm precisely when those thresholds are chosen optimally for the true underlying distributions, as noted in Section 2.
>
> ---
> **Q1:** Why the contextual-linear setting is defined on a $[-1/4,1/4]$ support?
>
> **A:** This choice is made purely for simplicity, avoiding the need to re‑derive some standard results under a different scaling. In fact, both our algorithm and analysis easily extend to any other $[-K,K]$ support setting by simply replacing every $[-1/4,-1/4]$-bounded argument with its $[-K,K]$-bounded analogue.
> We will clarify this in the revision to make it clear that $[-1/4,1/4]$ is for convenience and to make the paper concise by citing existing work where possible.
>
>
> ---
> **Q2:** Why the principle of optimism under uncertainty is helpful for improving the regret guarantee in this work?
>
> **A:** Our contribution here does not stem from the optimism under uncertainty principle, which Agarwal et al. (2024) already apply   yielding only a $\widetilde{O}(n\sqrt{T})$ regret bound.
> While we build on their framework, the improvement we obtain derives from taking different approach altogether wherein we  construct the *optimistic distribution* (shift probability mass based on Bernstein type bound), and more importantly, the improvement owes to our highly refined analysis.
> We highlight these differences in algorithm design and analysis in Section 3.1 and Section 3.2, respectively.
>
>
>
> ---
> **Q3:** What is the weak benchmark considered by Gergatsouli and Tzamos? Similarly, what is the optimal policy benchmark considered in Gatmiry et al.? What is the benchmark used in this work for both the non-contextual and contextual cases?
>
>
> **A:**
> Since Gergatsouli and Tzamos (2022) allow the  distributions of the boxes to be chosen adversarially across rounds, competing against the optimal strategy is intractable.
> Therefore to ensure sublinear regret, Gergatsouli and Tzamos (2022) introduce two weak benchmarks: (1) a non-adaptive benchmark that opens the same fixed set of boxes in every round, and (2) a partially-adaptive benchmark that may choose a different set in each round based on the algorithm’s past choices.
> On the other hand, Gatmiry et al., (2024) and our work (in both non-contextual and contextual settings) both use the *optimal strategy* as the benchmark. This is Weitzman's algorithm with  full knowledge of the product distribution at each round.
> We will make clarification in the next version.

---

> > ### Comment · Reviewer_HDSt · 2025-08-03
> >
> > Thanks for your clarifications. I think you can make it more clear that compared to prior work, your work introduces structural assumptions to make the comparison with optimal policy and sublinear guarantee achievable, if I understand it correctly. While I agree the technical contribution of your work is significant, I do think the writing needs improvement. That said, I'm happy to raise my score.

---

### Decision · Program_Chairs · 2025-09-17

**Decision:**

Accept (poster)

**Comment:**

The paper studies an online setting of the classic Pandora's box problem and improves the regret bound of prior work. Overall, reviewers are positive about the submission, and they think the paper has interesting technical contributions. I would also recommend authors to include some clarification discussions in the rebuttal to the final version of the paper.